# PIC v1.3: Comprehensive R package for permafrost indices computing with daily weather observations and atmospheric forcing over the Qinghai-Tibet Plateau

Lihui Luo[1], Zhongqiong Zhang[1], Wei Ma[1], Shuhua Yi[2], Yanli Zhuang[3]

[1]State Key Laboratory of Frozen Soils Engineering, Cold and Arid Regions Environmental and Engineering Research Institute, Chinese Academy of Sciences, Lanzhou, Gansu Province 730000, China

[2]State Key Laboratory of Cryospheric Sciences, Cold and Arid Regions Environmental and Engineering Research Institute, Chinese Academy of Sciences, Lanzhou, Gansu Province 730000, China

[3]Linze Inland River Basin Research Station, Key Laboratory of Inland River Basin Ecohydrology, Cold and Arid Regions Environmental and Engineering Research Institute, Chinese Academy of Sciences, Lanzhou, Gansu Province 730000, China

**Correspondence:** Lihui Luo (luolh@lzb.ac.cn)

**Abstract.** A permafrost indices computing (PIC v1.3) R package was developed that integrates meteorological observations, gridded meteorological datasets, soil databases, and field measurements to compute the factors or indices of permafrost and seasonal frozen soil. At present, 16 temperature/depth-related indices are integrated into the PIC v1.3 R package to estimate the possible trends of frozen soil in the Qinghai-Tibet Plateau (QTP). These indices include the mean annual air temperature (MAAT), mean annual ground surface temperature (MAGST), mean annual ground temperature (MAGT), seasonal thawing/freezing n factor ($n_t$/$n_f$), thawing/freezing degree-days for air and the ground surface ($DDT_a$/$DDT_s$/$DDF_a$/$DDF_s$), temperature at the top of permafrost (TTOP), active layer thickness (ALT), and maximum seasonal freeze depth. PIC v1.3 supports two computational modes, namely, the stations and regional calculations that enable statistical analysis and intuitive visualization of the time series and spatial simulations. Data sets of 52 weather stations and a central region of the QTP were prepared and simulated to evaluate the temporal-spatial trends of permafrost with the climate. More than 10 statistical methods and a sequential Mann-Kendall trend test were adopted to evaluate these indices in stations, and spatial methods were adopted to assess the spatial trends. Multiple visual manners were used to display the temporal and spatial variability of the stations and region. Simulation results show extensive permafrost degradation in the QTP, and the temporal-spatial trends of the

permafrost conditions in the QTP are close to those of previous studies. The transparency and repeatability of the PIC v1.3 package and its data can be used and extended to assess the impact of climate change on permafrost.

## 1 Introduction

Permafrost, which is soil, rock, or sediment with temperatures that have remained at or below 0 °C for at least two consecutive years, is a key component of the cryosphere. The upper layer in permafrost regions is called the active layer, and it undergoes seasonal freezing and thawing. Below this layer lies permafrost, the upper surface of which is called the upper permafrost limit or the permafrost table. Changes in permafrost can affect water and heat exchange, the carbon budget, and natural hazards with climate change. Permafrost occurs mostly in high latitudes and altitudes with long, cold winters and thick winter snow, e.g., the Arctic, Antarctica, Alaska, the Alps, Northern Russia, Northern Canada, Northern Mongolia, and the Qinghai-Tibet Plateau (QTP) (Riseborough et al., 2008; Yi et al., 2014a; Zhang et al., 2008a). Over half of the QTP is underlain by permafrost (Ran et al., 2012). The temperature in the QTP has increased by more than 0.25 °C per decade over the past 50 years (Li et al., 2010; Ran et al., 2018; Shen et al., 2015; Yao et al., 2007). Climate-induced warming of the near-surface atmospheric layer and a corresponding increase in ground temperatures will lead to substantial changes in the water and energy balance of regions underlain by permafrost (Hilbich et al., 2008). Such an increase in the temperature of the QTP can warm the ground through energy exchange at the surface and result in significant permafrost degradation. Understanding the distribution and changes of permafrost under the influence of climate change is necessary for infrastructure development, ecological and environmental assessment, and climate system modelling (Luo et al., 2017; Luo et al., 2012; Zhang et al., 2014).

Given the possibility of future climate warming, an evaluation of the magnitude of changes in the ground thermal regime has become desirable to assess the possible eco-environmental response and the impact on QTP infrastructure. Permafrost modelling maximizes quantitative analytical, numerical, or empirical methods, to predict the thermal condition of the ground in environments where permafrost may be present (Harris et al., 2009; Lewkowicz and Bonnaventure, 2008; Riseborough, 2011; Riseborough et al., 2008; Yi et al., 2014b; Zhang et al., 2008b). At present, dozens of different factors or indices are used to evaluate the characteristics and dynamics of permafrost presence or absence (Riseborough, 2011; Riseborough et al., 2008),

including the freezing/thawing index, mean annual air temperature (MAAT), mean annual ground temperature (MAGT), mean annual ground surface temperature (MAGST), temperature at the top of permafrost (TTOP), and the active layer thickness (ALT). Thereafter, the type and distribution of frozen soil can be classified in a variety of manners depending on the range and magnitude of these indices. For example, frozen soil can be divided into highly stable, stable, substable, transitional, unstable, and extremely unstable permafrost, as well as seasonal frozen soil that depends on the magnitude of MAGT (Chen et al., 2012; Ran et al., 2012). These indices can be used to evaluate and predict the temporal and spatial variation in the thermal response of permafrost to the changing climatic conditions and properties of Earth's surface and subsurface in one, two, or three dimensions (Juliussen and Humlum, 2007; Nelson et al., 1997; Riseborough et al., 2008; Wu et al., 2010; Zhang et al., 2005). Accordingly, successfully summarizing and categorizing a variety of frozen-soil indices requires permafrost modelling that concerns analytical, numerical, and empirical methodologies to compute the past and present conditions. The Stefan solution (Nelson et al., 1997), Kudryavtsev's approach (Kudryavtsev et al., 1977), the TTOP model (Smith and Riseborough, 1996), and the Geophysical Institute Permafrost Lab model (Romanovsky and Osterkamp, 1997; Sazonova and Romanovsky, 2003) are several important developments for permafrost modelling in recent years. Permafrost is a subsurface feature that is difficult to directly observe and map. These methods integrate the effects of air and ground temperatures, topography, vegetation, and soil properties to map permafrost spatially and explicitly (Gisnås et al., 2013; Jafarov et al., 2012; Zhang et al., 2014). Weather observation data, including air and soil temperatures at different depths, are the main inputs for single-point simulation, whereas the spatial and temporal resolution of the atmospheric forcing dataset is the main input data of permafrost spatial modelling. These permafrost indices consist mainly of temperature-related and depth-related indices. The temperature-related indices depict the status of air or land surface temperature in frozen-soil environments, whereas the depth-related indices reveal the status of the active layer. Preparing atmospheric forcing, snow depth and density, vegetation types, and soil class data sets from multi-source data fusion, particularly remote sensing and ground observation data is generally required for multi-dimensional permafrost simulation.

The transparency and repeatability of data, parameters, model codes, computational processes, simulation output, visualization, and statistical analysis is a fundamental principle of scientific research in Earth system modelling. At present, there is a lack of open source software, shared data and parameters for permafrost modelling in the QTP. Although many scientists in China

have field data and models on hand, their integration into a new open source model can facilitate the deepening of the discussion and unfolding of permafrost research on the QTP. Given the current situation of permafrost modelling in the QTP, a comprehensive R package of permafrost indices computing (PIC v1.3, doi: 10.5281/zenodo.1254848) was developed to compute permafrost and seasonal frozen-soil indices (Luo, 2018). The goal is to provide guidance for the future of highway

and high-speed railway design and construction in the QTP, as well as to further understand the effects of climate change on permafrost dynamics. Therefore, the proposed software integrates meteorological observations, gridded meteorological datasets, soil databases, field measurements, and permafrost modelling.

## 2 Package description

### 2.1 Overview

PIC v1.3 was developed in the R language and environment for statistical computing v. 3.3.3 and is distributed as open source software under the GNU-GPL 3.0 License (R Core Team, 2017). Therefore, the PIC v1.3 code can be modified as required to meet the needs of every user. The source code can be downloaded from the GitHub repository (https://github.com/iffylaw/PIC). The R package PIC v1.3 provides all the necessary functionality to perform the calculation, statistics, and drawing of permafrost indices with over 38 functions based on the user's specific requirements (see

Figure 2). The following packages are required in setting up PIC v1.3 (type library(PIC)): ggplot2 (Wickham et al., 2009), ggmap (Kahle and Wickham, 2013), RNetCDF (Michna and Woods, 2013), and animation (Xie, 2013). These packages are automatically added to the R users' library during installation. A data set that contains the daily weather observations, parameters, and information (i.e., from 1951 to 2010) of 52 weather stations in the QTP was bundled into this package. However, the regional data with the NetCDF format was placed in the GitHub repository. The data set variables excluded in

the calculation can also be used as reference or provide support to further develop PIC. These variables include wind speed, precipitation, evaporation, humidity, and soil temperature at different depths. PIC v1.3 was primarily designed to compute indices of permafrost and seasonal frozen soil from observations and forcing data. Therefore, the current stable version of the program (v 1.1) includes functionalities that cover temperature-related indices (i.e., MAAT, MAGST, and TTOP) and depth-

related indices (i.e., ALT and FD) that are commonly used in permafrost research. It is possible to evaluate the changes in frozen soil better by combining multiple indices for overall analysis.

## 2.2 Permafrost modelling

PIC v1.3 enables the calculation of the thawing/freezing degree-days for air and ground surface ($DDT_a$/$DDT_s$/$DDF_a$/$DDF_s$), MAAT, MAGST, MAGT, the seasonal thawing/freezing $n$ factor ($n_t$/$n_f$), TTOP, ALT, and the maximum seasonal freeze depth (FD). The permafrost and seasonal frozen-soil indices employing the following functions are illustrated. Table 1 describes most of them.

$A_s$ is the annual temperature amplitude at the ground surface, where $T_{max}$ and $T_{min}$ are the annual maximal and minimal temperatures, respectively. $A_s$ can be calculated as follows:

$$A_s = T_{max} - T_{min} \tag{1}$$

$L$ is the volumetric latent heat of fusion, $\rho$ is the dry density of soil, and $W$ is the water content of the soil in percentages.

$$L = \frac{80 \times \rho \times W}{100} \tag{2}$$

$DDT_a$ and $DDT_s$ are the sums of mean daily air and ground surface above temperatures 0 °C (Celsius degree-days), respectively. $DDF_a$ and $DDF_s$ are the sums of mean daily air and ground surface temperatures below 0°C (Celsius degree-days), respectively. Degree-days are usually used to describe the air and ground surface temperature intensity, where $T_a$ and $T_s$ are the air and ground temperatures, respectively, and $n$ is the number of days in a year (Juliussen and Humlum, 2007).

$$DDT_a = \sum_1^n T_a, \ T_a > 0 \tag{3}$$

$$DDF_a = \sum_1^n T_a, \ T_a < 0 \tag{4}$$

$$DDT_s = \sum_1^n T_s, \ T_s > 0 \tag{5}$$

$$DDF_s = \sum_1^n T_s, \ T_s < 0 \tag{6}$$

$P$ is assigned a value of 365 days as a default value. Local variations in vegetation, topography, and snow cover may result in several differences between MAGST and MAAT. MAAT and MAGST can be computed as follows:

$$MAAT = \frac{DDT_a - DDF_a}{P} \tag{7}$$

$$\text{MAGST} = \frac{DDT_s - DDF_s}{P} \tag{8}$$

MAGT is defined as the soil temperature at the depth of zero annual temperature change. $T_{z,t}$ is the ground temperature at any time $t$ and depth $z$ below a ground surface. MAGT is often found at depths from 10 m to 15 m over the QTP (Wu and Zhang, 2010), Here, we take the $z$ value of 15 metres as default value, user can change the depth $z$. MAGT can be computed (Juliussen and Humlum, 2007; Riseborough et al., 2008) as follows:

$$\text{T}_{z,t} = \overline{T_a} + A_s \times e^{-z \times \sqrt{\pi/\alpha P}} \times \sin\left(\frac{2\pi t}{P} - z \times \sqrt{\pi/\alpha P}\right) \tag{9}$$

$$\text{MAGT} = \overline{\text{T}_{z,t}}, \ z \cong 15 \ \& \ t = 86400 \tag{10}$$

The seasonal thawing/freezing $n$ factor ($n_t/n_f$) relates thawing and freezing degree-days ($DDT_a/DDT_s/DDF_a/DDF_s$) in seasonal air temperature to ground surface temperatures. $n_t$ and $n_f$ can be computed (Riseborough et al., 2008) as follows:

$$n_t = \frac{DDT_s}{DDT_a} \tag{11}$$

$$n_f = \frac{DDF_s}{DDF_a} \tag{12}$$

TTOP indicates average temperatures at the top of the permafrost. The active layer is defined as the layer of ground subject to annual thawing and freezing underlain by permafrost. ALT refers to the maximum thawing depth of the active layer. Two methods serve the same purpose when computing TTOP and ALT. The subscripts $S$ and $K$ stand for the Smith and Kudryavtsev functions (Kudryavtsev et al., 1977; Smith and Riseborough, 1996), respectively.

$$\text{TTOP}_S = \frac{n_t \times \lambda_t \times DDT_a - n_f \times \lambda_f \times DDT_s}{\lambda_f \times P} \tag{13}$$

$$\text{TTOP}_K = \frac{0.5 \times MAGST \times (\lambda_t + \lambda_f) + A_s \times \frac{\lambda_f - \lambda_t}{\pi} \times \left[\frac{MAGST}{A_S} \times arcsin\frac{MAGST}{A_S} + \sqrt{1 - \frac{\pi^2}{A_S^2}}\right]}{\lambda^*} \tag{14}$$

$$\lambda^* = \begin{cases} \lambda_f, if \ numerator < 0 \\ \lambda_t, if \ numerator > 0 \end{cases} \tag{15}$$

The maximum thawing depth or ALT uses the Stefan and Kudryavtsev functions (Kudryavtsev et al., 1977; Riseborough et al., 2008), where $L$ is the latent heat of fusion for ice ($3.34 \times 10^5$ J/kg).

$$\text{ALT}_S = \sqrt{\frac{2 \times \lambda_t \times DDT_a}{L \times \rho \times (W - W_u)}} \tag{16}$$

$$A_z = \frac{A_s - T_z}{ln\left[\frac{A_s + L/2 \times C_T}{T_z + L/2 \times C_T}\right]} - \frac{L}{2 \times C_T} \tag{17}$$

$$Z_c = \frac{2 \times (A_s - T_z) \times \sqrt{\frac{(\lambda_f + \lambda_t) \times P_{sn} \times C_T}{2 \times \pi}}}{2 \times A_z \times C_T + L} \tag{18}$$

$$ALT_K = \frac{2 \times (A_s - TTOP_K) \times \sqrt{\frac{(\lambda_f + \lambda_t) \times P_{sn} \times C_T}{2 \times \pi}} + \frac{(2 \times A_z \times C_T \times Z_c - L \times Z_c) \times L \times \sqrt{\frac{(\lambda_f + \lambda_t) \times P_{sn}}{2 \times \pi \times C_T}}}{2 \times A_z \times C_T \times Z_c + L \times Z_c + (2 \times A_z \times C_T + L) \times \sqrt{\frac{(\lambda_f + \lambda_t) \times P_{sn}}{2 \times \pi \times C_T}}}}{2 \times A_z \times C_T + L} \tag{19}$$

*Freeze_depth$_s$* is the maximum seasonal freezing depth that uses the Stefan function, which can be computed as follows:

$$\text{Freeze\_depth}_S = \sqrt{\frac{2 \times \lambda_t \times DDF_a}{L \times \rho \times (W - W_u)}} \tag{20}$$

## 2.3 Statistical methods

Statistical analysis can facilitate evaluation of the trend and the overall modelling performance. In particular, each statistic has strengths and weaknesses. Thus, we adopted over 10 statistical methods to evaluate these indices in station computing for time series data. The quantitative statistics include the slope, *y*-intercept, Pearson's correlation coefficient (*R*), coefficient of determination ($R^2$), root mean square error (RMSE), standard deviation (SD), ratio of scatter (RS), normalized RMSE (NRMSE), Nash-Sutchliffe efficiency (NSE), RMSE-observations standard deviation ratio (RSR), percent bias (PBIAS), normalized average error (NAE), variance ratio (VR), and index of agreement (*D*) (Jafarov et al., 2012; Legates and McCabe, 1999). The sequential Mann-Kendall (MK) trend test was used to statistically assess whether there was a shift in trends of the climate factors and permafrost indices (Fraile, 1993). The original MK trend test can be calculated as follows:

$$S = \sum_{i=1}^{n-1} \sum_{j=i+1}^{n} sign(x_j - x_i), (i = 2,3,4 \dots n) \tag{21}$$

$$sign(x_j - x_i) = \begin{cases} 1 \ if \ x_j - x_i > 0 \\ 0 \ if \ x_j - x_i = 0 \\ -1 \ if \ x_j - x_i < 0 \end{cases} \tag{22}$$

Two sequential series $u_i$ values can be calculated as follows:

$$u_i = \frac{S_i - E(S_i)}{\sqrt{Var(S_i)}}, (i = 1,2,3 \dots n) \tag{23}$$

The two series for the MK trend test, a progressive and a backward, were set up. If they cross each other and diverge beyond

a specific threshold value and exceed the confidence level of 95%, then there is a statistically significant trend shift point.

The spatial trend can also be calculated to evaluate regional computing for temporal-spatial data through the function below. The *index* represents one permafrost index, *n* represents the sequential years, and $index_i$ is the index value in year *i*. Taking ALT as an example, a positive trend means that ALT was increasing, thereby indicating that permafrost degradation has intensified; a negative value means that ALT was decreasing, thereby indicating that permafrost degradation has a certain inhibition; and a zero trend suggests a lack of change (Chen et al., 2014; Stow et al., 2003).

$$\text{Trend} = \frac{n \times \sum_{i=1}^{n} i \times index_i - \sum_{i=1}^{n} i \times \sum_{i=1}^{n} index_i}{n \times \sum_{i=1}^{n} i^2 - (\sum_{i=1}^{n} i)^2} \tag{24}$$

## 3 Data and parameters

### 3.1 Daily weather observations

Table 2 shows detailed information of the data and parameters. Meteorological data were obtained from the China Meteorological Administration (CMA, http://www.cma.gov.cn/), particularly from permanent meteorological stations across the QTP (Figure 1). A total of 52 weather stations with daily meteorological records (i.e., from 1951 to 2010) were selected, including the daily mean, maximum (max) and minimum (min) air temperatures, wind speed, observed and corrected precipitation, evaporation, air humidity, atmospheric pressure, sunshine duration, daily mean, max and min ground surface temperatures, and soil temperature at different depths (i.e., 5, 10, 15, 20, 40, 50, 80, 160, and 320 cm). These data have been corrected under specifications for surface meteorological observation and CMA quality control. Daily weather observations are used as the input data for the PIC v1.3 station calculation.

### 3.2 Atmospheric forcing dataset

The Qinghai-Tibet Engineering Corridor (QTEC), located at the centre of the QTP, was selected in preparing the atmospheric forcing data. Global Land Data Assimilation System (GLDAS, https://ldas.gsfc.nasa.gov) and the weather station data of the surrounding QTEC were merged through spatial interpolation and offset correction to produce a new data set for 1980 to 2010 with a daily 0.1° temporal-spatial resolution (Luo et al., 2018). An atmospheric forcing dataset was used as the input data for the PIC v1.3 regional calculation.

### 3.3 Parameters

The parameters for the ground conditions were based on soil property data and field observations. The parameter data have two sets: one for weather stations and another for the QTEC region. The Harmonized World Soil Database (HWSD, version 1.21) provides information on soil parameters that are available for evaluating soil thermal conductivity with field observations and can be used as input parameters to the PIC v1.3 package (Bicheron et al., 2008; Nachtergaele et al., 2009). The thermal conductivity of ground in a thawed/frozen state, $\lambda_t$ and $\lambda_f$, can be computed through the joint parameterization scheme of the Johansen method (Johansen, 1977) and Luo parameterization (Luo et al., 2009):

$$\lambda_{dry} = \frac{0.135 \times \rho + 64.7}{2700 - 0.947 \times \rho} \tag{25}$$

$$\lambda_s = \lambda_q{}^q \times \lambda_o{}^{1-q} \tag{26}$$

$$\lambda_{sat} = \lambda_s{}^{1-\theta_s} \times \lambda_w{}^{\theta_s} \tag{27}$$

$$S_r = \frac{\theta}{\theta_s} \tag{28}$$

$$K_{et} = \frac{K_t \times S_r}{1 + (K_t - 1) \times S_r} \tag{29}$$

$$K_{ef} = \frac{K_f \times S_r}{1 + (K_f - 1) \times S_r} \tag{30}$$

$$\lambda_t = (\lambda_{sat} - \lambda_{dry})K_{et} + \lambda_{dry} \tag{31}$$

$$\lambda_f = (\lambda_{sat} - \lambda_{dry})K_{ef} + \lambda_{dry} \tag{32}$$

where the soil thermal conductivity of dry soil $\lambda_{dry}$ depends on dry bulk density $\rho$, the thermal conductivity of soil solids $\lambda_s$ varies with the gravel content q, $\lambda_q$ is the thermal conductivity of quartz (7.7 W m$^{-1}$ K$^{-1}$), $\lambda_o$ is the thermal conductivity of other minerals (2.0 W m$^{-1}$ K$^{-1}$), and q is the gravel content in the soil. The saturated soil thermal conductivity $\lambda_{sat}$ depends on the thermal conductivity of soil solids $\lambda_s$, liquid water $\lambda_w$ (0.594 W m$^{-1}$ K$^{-1}$), and the soil saturated water content $\theta_s$. The degree of saturation $S_r$ is a function of the soil water content, $\theta$ and soil saturated water content, $\theta_s$. The Kersten numbers in the thawed/frozen state, $K_{et}$ and $K_{ef}$, are two functions of the degree of saturation $S_r$, and K values in the thawed/frozen state, $K_t$ and $K_f$; $\rho$, q and $\theta_s$ come from the T_BULK_DENSITY, T_GRAVEL, and T_BS fields of the HWSD.

The volumetric heat capacity during thawing, $C_T$, is given as :

$$C_T = (C_s + \theta \times C_w) \times \rho \tag{33}$$

Where $C_w$ is specific heat of liquid water (4.18 kJ kg$^{-3}$ K$^{-1}$), $C_s$ is soil specific heat capacity. $\theta$, $C_s$, $K_t$ and $K_f$ in different soil textures can be found in Table 3, these four parameters are empirical parameters used to explain different soil texture types based on soil texture, thermal conductivities and specific heat capacity derived from soil sampling along the QTEC. Figure 3 shows these input spatial parameters over the QTP.

## 4 Implementation

PIC v1.3 supports two computational modes: the station and regional calculations that enable statistical analysis and visual displays of the time series and spatial simulations. The regional calculation adopts GIS approaches to compute each spatial grid. PIC v1.3 was initially developed to address the immediate need for a reliable and easy-to-use program for estimating temporal-spatial changes in frozen QTP soil. Thus, the workflow is comprised of deliberately simplified steps throughout the entire process. Once PIC v1.3 is installed, the workflow of the weather observations is considerably straightforward: (1) an index of a weather station for one year or multiple years is calculated, (2) an index of 52 weather stations from 1951 to 2010 is calculated, and (3) an index of all stations or permafrost stations from 1951 to 2010 is drawn through a curve and spatial visualization. Step (1) is an optional step. The forcing data workflow has only two steps: (1) a total of 4 indices from 1980 to 2010 are calculated, including MAAT, $DDT_a$, $DDF_a$, and ALT and (2) the spatial statistics and visualization of these 4 indices are drawn.

Several examples of PIC v1.3 use and application are presented here. This section highlights several significant features of the package in terms of specific functions, including station and regional calculation, statistics, and visualization. However, PIC v1.3 includes numerous illustrations from the literature and possible detailed analyses. PIC v1.3 has built-in station data. The data set comprises two tables (data frame), namely, QTP_ATM for daily weather observations and Station_Info for information and parameters from each station. Users can modify or adjust the parameters in the Station_Info and use the data and parameters. Additional examples can be referenced in the GitHub repository (https://github.com/iffylaw/PIC/blob/master/ Examples.R).

## 4.1 Station calculation

We can use different functions in the R console to perform the calculations based on the selected method. For example, if a user wants to obtain a MAAT value for a certain station year, he/she can enter the following command. TempName and data are optional in the MAAT function.

```
MAAT (Year = 1980, TempName = "Temperature", data = QTP_ATM, SID = 52908)
```

A user can also obtain the MAAT values for a specified period of years in a station.

```
MAAT (Year = 1980:2010, TempName = "Temperature," data = QTP_ATM, SID = 52908)
```

The "Year" option can be assigned to a number and sequence. The other temperature/depth-related indices can also use the two inputs for the "Year" option. A user can obtain the values of all stations for an index. The "VarName" option can be equal to the function name in the Com_Indices_QTP function. The results can also be saved to a CSV file with column/row names. The case of the input VarName is supported.

```
Com_Indices_QTP (VarName = "MAAT")
```

Given that the freezing/thawing index can be divided into freezing/thawing degree-days of the air and ground surface, the VarName option should add "_air" or "_ground" at the ends of the Freezing_index and Thawing_index. However, the abbreviation can also be utilized as the option input. The "Thawing_index_air" and "ta" are the same.

```
Com_Indices_QTP (VarName = "Thawing_index_air")
Com_Indices_QTP (VarName = "ta")
```

After the TTOP indices are computed, the stations that may have permafrost should be determined. The Exist_Permafrost function can determine and map the stations where permafrost exists. The probability of permafrost occurrence and most likely permafrost conditions are determined from the computing results of the Exist_Permafrost function (see Figure 4).

```
TTOP_S_QTP <- Com_Indices_QTP (VarName = "TTOP_Smith")
TTOP_K_QTP <- Com_Indices_QTP (VarName = "TTOP_Kudryavtsev")
Exist_Permafrost (plot = "yes")
```

The QTP measurements have constantly been difficult. The data set has several null and anomalous values, as well as leading

to a few anomalous values in computing the indices. Accordingly, these outlier values should be processed. The Outlier_Process function seeks the outlier values and sets them to null thereafter, which is an option because abnormal values have been processed in the Com_Indices_QTP.

```
Outlier_Process (MAAT_QTP[,1:stations])
```

## 4.2 Regional calculation

A total of four indices, including MAAT, DDF$_a$, DDT$_a$ and ALT, can be computed with the atmospheric forcing data set in PIC v1.3. This package supports NetCDF format data; thus, it reads and writes a NetCDF file to support regional computing. The input NetCDF files require a few forcing and parameter data. After the calculations, a user can compute the spatial statistics and draw the index changes through a GIF animation (see Sections 4.3 and 4.4).

```
Spatial_Pic (NetCDFName = "PIC_indices.nc", StartYear = 1980, EndYear = 2010)
```

## 4.3 Statistics

The stat function contains all the statistical methods for station calculation. PIC v1.3 provides two calculations for computing the statistical values of all stations using Com_Stats_QTP: (1) the indices that vary with changing years and (2) the comparison of the same two indices for different computational methods. Options ind1 and ind2 were used; however, ind2 can be disregarded when computing the statistical values between a single datum and years.

```
Com_Stats_QTP (ind1 = MAAT_QTP)
```

TTOP and ALT were calculated utilizing two different functions, so these two indices should be compared. For example, the
two TTOP values for all QTP stations are compared. A user can assign ind1 and ind2 to compute the ALT statistical values between the Stefan and Kudryavtsev functions. Thereafter, the statistical values are saved to the CSV file when executing the Com_Stats_QTP function. Table 4 shows all the statistical values of the selected stations.

```
Com_Stats_QTP (ind1 = TTOP_S_QTP, ind2 = TTOP_K_QTP)
Com_Stats_QTP (ind1 = ALT_S_QTP, ind2 = ALT_K_QTP)
```

A spatial trend can also be computed using the Spatial_Stat function after the regional calculation. The function simultaneously

saves the spatial trend of the five indices into the NetCDF file. In addition, the function draws the animation of the spatial trend (see Section 4.4).

Spatial_Stat ("PIC_indices.nc", "ALT")

## 4.4 Visualization

Station visualization can be produced by Plot_TTOP_ALT and Plot_3M. The Plot_TTOP_ALT function plots two TTOP or

5    two ALT indices in a figure for all stations or stations with permafrost. VarName has the "TTOP" and "ALT" options, whereas SID has the "permafrost" and "all" options. The Plot_3M function draws the MAAT, MAGST, and MAGT indices. The two functions plot only the stations where permafrost exists when SID = "permafrost."

Plot_TTOP_ALT (VarName = "TTOP", SID = "permafrost")

Plot_TTOP_ALT (VarName = "ALT", SID = "permafrost")

Plot_3M(SID = "permafrost")

The other approach of "ggplot2" was adopted to visualize the region (see Figure 5).

ggplot_Pic (Type = "TTOP", SID = "permafrost")

The indices that change over time can also be plotted through a GIF animation that uses Map_Pic (Figure 6).

Map_Pic (VarName = "TTOP_S")

Map_Pic (VarName = "TTOP_K")

10    The input and output of the regional calculation can be drawn using the Netcdf_Multiplot function (see Figure 7), which uses animation to display the values. The spatial trend can also be drawn in the Spatial_Stat apart from calculating the spatial statistics. This function draws all four indices when "VarName" has no input (see Figure 8).

Netcdf_Multiplot (NetCDFName = "PIC_indices.nc", VarName = "ALT")

Netcdf_Animation (NetCDFName = "PIC_indices.nc", VarName = "ALT")

Spatial_Stat ("PIC_indices.nc")

## 5 Discussion

### 5.1 PIC performance

This study proposes permafrost modelling to compute the changes in the active layer and permafrost with the climate, and this considers station and regional modelling over the QTP. We apply the two approaches to 52 weather stations and a central region of the QTP. The PIC v1.3 simulation results using the Exist_Permafrost function show that permafrost was detected at 12 of the 52 observation stations (Figure 4). The permafrost areas began to shrink from the southern and northern parts to the central QTEC region (Figure 7). The permafrost, whether in permafrost stations or QTEC, continued to thaw with increasing ALT, low surface offset and thermal offset, and high MAAT, MAGST, MAGT, and TTOP for most areas of QTP.

PIC v1.3 computes and maps the temporal dynamics and spatial distribution of permafrost in the stations and region. The regional modelling underwent more challenges than the stations' input data and parameters. The station calculation can estimate the long-term temporal trend of permafrost dynamics, whereas the regional calculation can estimate the temporal-spatial trend. In addition, the simulated TTOP and ALT using the Stefan and Smith functions are higher than the Kudryavtsev function. Although the overall trend of TTOP and ALT are coincidental, the two different computational methods can be combined to simulate their variation. Furthermore, 16 indices can be collectively employed for a comprehensive analysis. Moreover, the station and regional modelling can be integrated to evaluate the temporal-spatial evolution of permafrost in the QTP. In particular, the station modelling can be applied to validate the simulated results of the region. Moreover, the regional calculation can extend from QTEC to the entire QTP and even the other permafrost regions.

The "for" loop is discarded, whereas the "apply" functions are used extensively to significantly lower the computation time. PIC v1.3 was run natively as a single process in the Windows 7 Operating system. The calculations were performed independently through RStudio Desktop v1.1 software (RStudio, Inc., USA). The utilized processor is an Intel Core i7-2600 CPU 3.40 GHz, and the available memory is 32 GB. The current regional calculation takes only approximately 11 s. Apart from the Kudryavtsev model that requires considerable computation time (i.e., approximately 5 min), the station calculation also exhibited an improved efficiency. Therefore, PIC v1.3 can be considered an efficient R package.

Climate change indicates a pronounced warming and permafrost degradation in the QTP with active layer deepening (Chen et

al., 2013; Cheng and Wu, 2007b; Wu and Zhang, 2010; Wu et al., 2010), and both the simulation of stations and the region in PIC v1.3 also show widespread permafrost degradation (Figures 4-8). Meanwhile, as shown in Figures 7 & 8, the permafrost in the QTEC also continued to thaw, with the ALT growing. The QTEC is the most accessible area of the QTP. Most boreholes were drilled in the QTEC to monitor changes of permafrost conditions, and this monitoring data provides support for model performance evaluation. Meanwhile, ALT was widely used, so we adopted the permafrost index to estimate PIC v1.3 simulation performance. The simulated PIC v1.3 ALT and previous literature in the QTEC are compared in Table 5. The increasing rate of ALT averaged 0.50-7.50 cm yr$^{-1}$. The rate during the 1990s to 2010s was greater at more than 4.00 cm yr$^{-1}$, than during 1980 to the 1990s, at approximately 2.00 cm yr$^{-1}$. Though both the observed and the simulated ALT and its variation in different locations of the QTEC were still relatively large, the ALT trend in PIC v1.3 was close to the observations and simulation in the QTEC. In recent decades, the permafrost thaw rate has increased significantly. The majority of observed ALT and its trend along the QTH and QTR were greater than the simulated grid ALT of PIC v1.3, mainly because the observation sites are near these engineering facilities. These comparative analyses suggest that the temporal-spatial trends of permafrost conditions in the QTEC using PIC v1.3 were consistent with previous studies. More importantly, the difference between simulation results highlights the importance of transparency and reusability of models, data, parameters, simulation results and so on.

## 5.2 Advantages

Previous studies on the QTP (1) used one or two indices, such as MAAT and MAGST, to evaluate the permafrost changes (Yang et al., 2010), (2) constructed a regression analysis method through the relationship between MAGT and elevation, latitude, and slope-aspects that presented a static permafrost distribution (Lu et al., 2013; Nan, 2005), (3) did not share the model data and codes; hence, other researchers could not validate their results and conduct further research (McNutt, 2014). Compared with the previous permafrost modelling on the QTP, PIC v1.3 is considerably open, easy, intuitive, and reproducible for integrating data and most of the temperature/depth-related indices. The PIC v1.3 function supports the computation of multiple indices and different time periods, and the encoding mode is reusable and universal. This package can also be easily adopted to intuitively display the changes in the active layer and permafrost, as well as assess the impact of climate change.

The PIC v1.3 workflow is extremely simple and requires only one or two steps to obtain the simulated results and visual images. All running examples, data and code can be obtained from the GitHub repository. However, the permafrost modelling integrates a gridded meteorological dataset, soil database, weather and field observations, parameters, and multiple functions and models, supporting dynamic parameter changes such as vegetation and ground condition changes. Over 50 QTP weather stations were introduced, and they can partially resolve the spatial change of the permafrost area. The QTEC region is an example of spatial modelling that classifies land cover and topographic features to determine the spatial input parameters. Spatial modelling also uses the temporal-spatial data to provide spatially detailed information of the active layer and permafrost. The static/dynamic maps and statistical values of these indices can facilitate the understanding of the current condition of the near-surface permafrost and identify stations and ranges at high risk of permafrost thawing with the changing climate and human activities. Permafrost thawing causes significant changes in the environment and characteristics of frozen-soil engineering (Larsen et al., 2008; Niu et al., 2016). A comprehensive assessment of permafrost can provide guidance regarding the future of highways and high-speed railway systems in the QTP.

### 5.3 Limitations and uncertainties

PIC v1.3 was developed with numerous indices, as well as support station and regional simulation. PIC v1.3 can be used to estimate the frozen soil status and possible changes over the QTP by calculating permafrost indices. This package has many engineering applications and can be used to assess the impact of climate change on permafrost. Moreover, it provides observational data and a comprehensive analysis ability for multiple indices. The probability of permafrost occurrence and the most likely permafrost conditions are determined by computing the 16 indices. Although PIC v1.3 quantitatively integrates most of them based on previous studies (Jafarov et al., 2012; Nelson et al., 1997; Riseborough et al., 2008; Smith and Riseborough, 2010; Wu et al., 2010; Zhang et al., 2005; Zhang et al., 2014), it still has several limitations and uncertainties. First, the regional calculation is one-dimensional and assumes that each grid cell is uniform without water-heat exchange. Second, the heterogeneity in ground conditions of the QTP also brings along uncertainties of parameter preparation. Third, soil moisture at different depths affects the thermal conductivity and thermal capacity of the soil (Shanley and Chalmers, 1999; Yi et al., 2007). Thus, the soil input parameters should be dynamically changed. Lastly, climate forcing has several

uncertainties (Zhang et al., 2014), including input air and ground temperatures (i.e., the quality of the ground temperature in the QTP is currently unreliable). Thus, the regional calculation supports fewer indices than the station calculation. These deficiencies can be significant for the permafrost dynamics with environmental evolution.

## 6. Conclusions

An R package PIC v1.3 that computes the temperature/depth-related permafrost indices with daily weather observations and atmospheric forcing has been developed. This package is open source software and can be easily used with input data and parameters that users can customize. A total of 16 permafrost indices for stations and the region are developed, and datasets of 52 weather stations and a central region of the QTP were prepared. Permafrost modelling and data are integrated into the PIC v1.3 R package to simulate the temporal-spatial trends of permafrost with the climate estimate and estimate the status of the

active layer and permafrost in the QTP. The current functionalities also include time-series statistics, spatial statistics, and visualization. Multiple visual manners display the temporal and spatial variability of the stations and the region. The package produces high-quality graphics that illustrate the status of frozen soil and may be used for subsequent publication in scientific journals and reports. The simulated PIC v1.3 results generally indicate that the temporal-spatial trends of permafrost conditions essentially agree with previously published studies. The transparency and repeatability of the PIC v1.3 package and its data

can be used to assess the impact of climate change on permafrost. Additional features may be implemented in future releases of PIC to broaden its application range. In the future, the observational data of the active layer will be integrated into the PIC datasets, and the simulation results will be compared with it. PIC v1.3 will also be used to predict the future state of permafrost by utilizing projected climate forcing and scenarios. Additional functions and models will be absorbed into PIC to improve the simulation and perform comparative analyses with other functions and models. Parallel computation will be added to improve

the computation efficiency. The key impact that PIC v1.3 is expected to provide to the open community is an increase in consistency within, and comparability among, studies. Furthermore, we encourage contributions from other scientists and developers, including suggestions and assistance, to modify and improve the proposed PIC v1.3.

**Code availability**

The PIC v1.3 code that supports the findings of this study is stored in the GitHub repository (https://github.com/iffylaw/PIC).

**Data availability**

The data are included in the Supplement files or GitHub repository.

**Competing interests**

The authors declare no competing financial interests.

**Acknowledgements**

This research was supported by the National Natural Science Foundation of China (41301508, 41630636, 41771074). We would like to express our gratitude to the editor and the two anonymous reviewers for their insightful comments and suggestions that improved this paper.

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

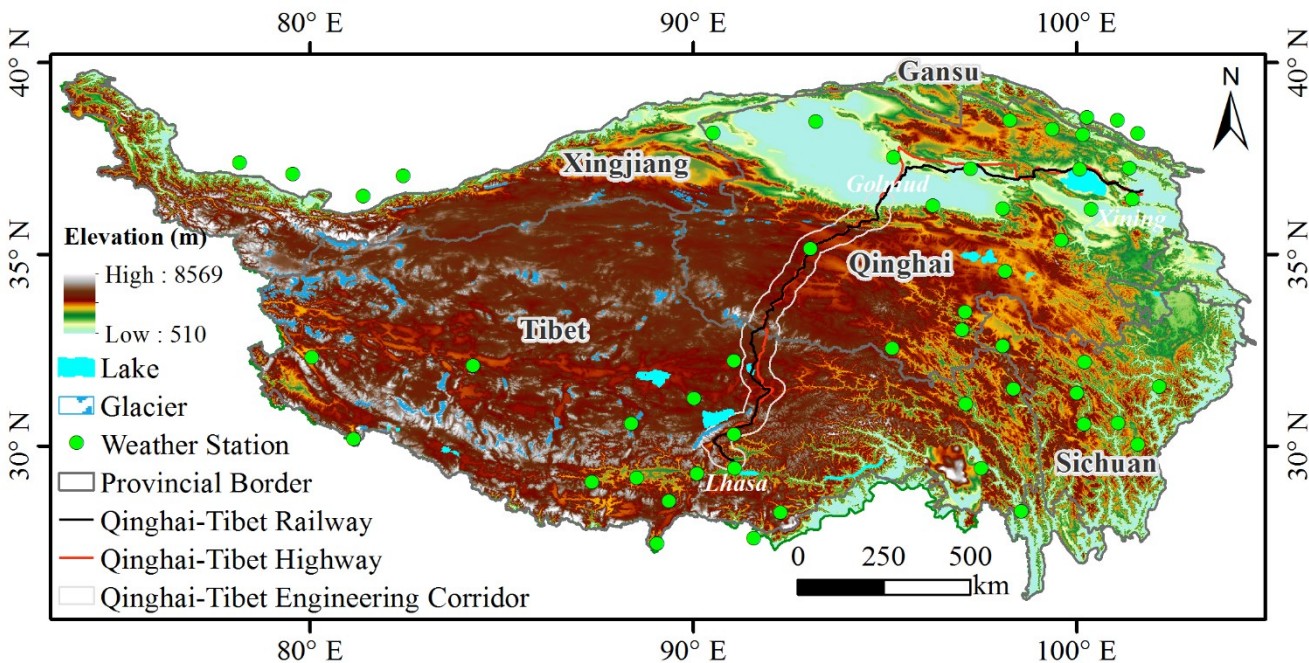

**Figure 1.** Map of the data location over the QTP.

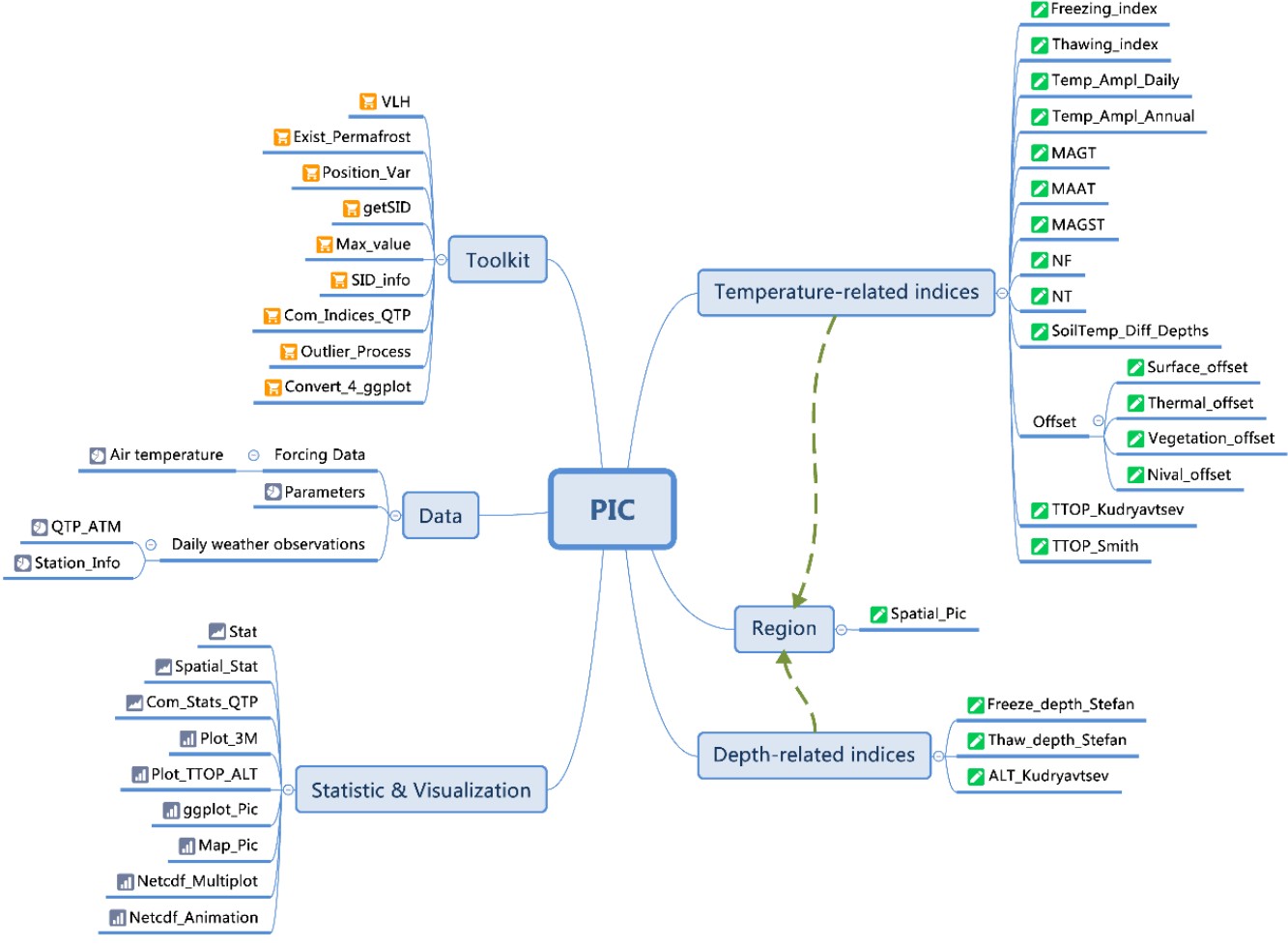

**Figure 2.** Mind map of the R package PIC v1.3.

**Table 1.** Most important user functions in the R package PIC v1.3. The equation column of this table corresponds to the equation in Section 2.

| R function | Equation | Description | Unit |
|---|---|---|---|
| **Temperature-related indices** | | | |
| Freezing_index | (4,6) | Freezing degree-days for air and ground | °C day |
| Thawing_index | (3,5) | Thawing degree-days for air and ground | °C day |
| MAAT | (7) | Mean annual air temperature | °C |
| MAGST | (8) | Mean annual ground surface temperature (5 cm) | °C |
| MAGT | (10) | Mean annual ground temperature (at 15 m) | °C |
| NT | (11) | Thawing n factor | |
| NF | (12) | Freezing n factor | |

| | | | |
|---|---|---|---|
| Surface_Offset | | The difference between the MAGST and MAAT | °C |
| Thermal_Offset | | The difference between the TTOP and MAGST | °C |
| Vegetation_Offset | | The second term (Surface_Offset) is negative and represents the reduction in MAGST due to vegetation effects in summer (vegetation offset) | °C |
| Nival_Offset | | The first term (Surface_Offset) on the right-hand-side is positive and represents the elevation of MAGST over MAAT due to the insulating effect of winter snow cover (nival offset) | °C |
| TTOP_Smith | (13) | The temperature at the top of the permafrost using Smith & Riseborough function | °C |
| TTOP_Kudryavtsev | (14) | The temperature at the top of the permafrost using Kudryavtsev function | °C |
| **Depth-related indices** | | | |
| Freeze_depth_Stefan | (20) | Maximum freezing depth using Stefan function | m |
| Thaw_depth_Stefan | (16) | Active layer thickness using Stefan function | m |
| ALT_Kudryavtsev | (19) | Active layer thickness (ALT) or maximum thawing depth using Kudryavtsev function | m |
| **Region** | | | |
| Spatial_Pic | (3,4,7,16) | Spatial changes with MAAT, $DDT_a$, $DDF_a$ and ALT | m |
| **Toolkit** | | | |
| Com_Indices_QTP | | Computing all indices for all stations of the QTP | |
| Outlier_Process | | Process the abnormal value | |
| VLH | (2) | Computing volumetric latent heat of fusion | J/m$^3$ |
| Convert_4_ggplot | | Convert the values of TTOP & ALT to one column | |
| Exist_Permafrost | | To determine the stations where permafrost exist by TTOP values | |
| **Statistic** | | | |
| Stat | (21,22,23) | Statistical functions with 10 more methods | |
| Spatial_Stat | (24) | Spatial statistical method, just for spatial trend | |
| Com_Stats_QTP | | Computing the statistical values for one or both of these indices | |
| **Visualization** | | | |
| Plot_3M | | Plot MAAT, MAGST, and MAGT for all stations or a single station | |
| Plot_TTOP_ALT | | Plot TTOP and ALT for all stations or a station | |

| | Plot multiple indices for all stations or a single station using ggplot2 |
|---|---|
| ggplot_Pic | Plot multiple indices for all stations or a single station using ggplot2 |
| Map_Pic | Plot multiple indices for all stations or a single station using ggmap |
| Netcdf_Multiplot | Regional visualization of NetCDF with multiple plots |
| Netcdf_Animation | Regional animation of NetCDF |

**Table 2.** Input data and parameters.

| Variables | Meaning | Unit |
|---|---|---|
| Temperature | Daily mean air temperature | °C |
| Tmax | Daily maximum air temperature | °C |
| Tmin | Daily Minimum air temperature | °C |
| GT | Daily mean ground temperature in 0 cm | °C |
| GT_0_MAX | Daily maximum ground temperature at 0 cm | °C |
| GT_0_MIN | Daily minimum ground temperature at 0 cm | °C |
| temp | Spatial daily mean air temperature | °C |
| $\lambda_t$ | Thermal conductivity of ground in thawed state | W/m K |
| $\lambda_f$ | Thermal conductivity of ground in frozen state | W/m K |
| L | Latent heat of fusion | J/m$^3$ |
| $\rho$ | Dry bulk density | kg/m$^3$ |
| W | Soil water content in thawed state | % |
| $W_u$ | Soil unfrozen water content in frozen state | % |
| $P_{sn}$ | period of the temperature wave, adjusted for snow melt | s |
| $C_T$ | volumetric heat capacity during thawing | kJ/m$^3$ K |

**Table 3**. Parameters of thermal conductivity in the thawed/frozen state. The UADS Code came from soil texture classification
of United States Department of Agriculture (USDA). The Qinghai-Tibet Plateau does not have the 1 and 8 of soil classification
codes. θ: soil water content; $K_t$: K value in thawed state; $K_f$: K value in frozen state; $C_s$: specific heat capacity in thawed stat
(kJ/kg K).

| USDA Code | Soil Texture | $\theta$ | $K_t$ | $K_f$ | $C_s$ |
|---|---|---|---|---|---|

| 1 | clay(heavy) | 0.17 | 1.90 | 0.85 | 1.00 |
|---|---|---|---|---|---|
| 2 | silty clay | 0.17 | 1.90 | 0.85 | 1.00 |
| 3 | clay (light) | 0.17 | 1.90 | 0.85 | 0.92 |
| 4 | silty clay loam | 0.17 | 1.90 | 0.85 | 0.92 |
| 5 | clay loam | 0.17 | 1.90 | 0.85 | 0.92 |
| 6 | silt | 0.17 | 1.90 | 0.85 | 0.87 |
| 7 | silt loam | 0.17 | 1.90 | 0.85 | 0.87 |
| 8 | sandy clay | 0.15 | 3.55 | 0.85 | 0.84 |
| 9 | loam | 0.15 | 3.55 | 0.95 | 0.84 |
| 10 | sandy clay loam | 0.15 | 3.55 | 0.95 | 0.84 |
| 11 | sandy loam | 0.15 | 3.55 | 0.95 | 0.84 |
| 12 | loamy sand | 0.06 | 4.60 | 1.70 | 0.79 |
| 13 | sand | 0.06 | 4.60 | 1.70 | 0.79 |

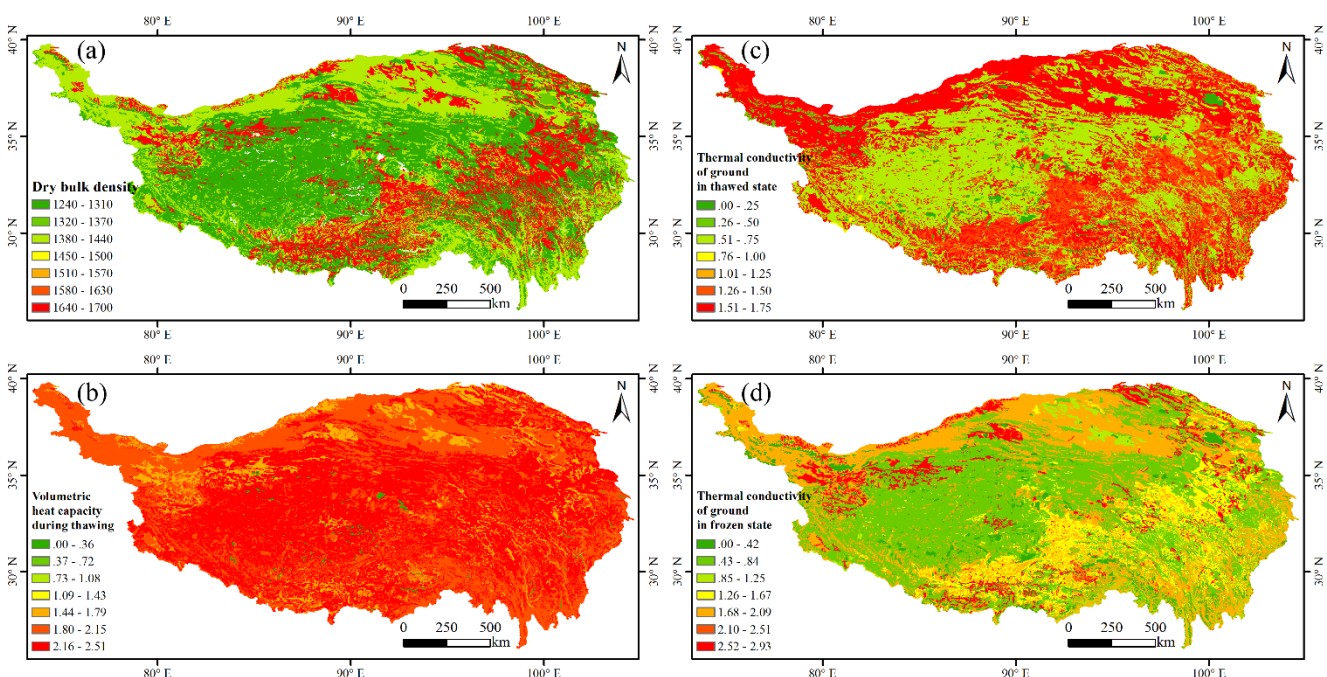

**Figure 3.** Spatial parameters for PIC v1.3 over the Qinghai-Tibet Plateau. (a) dry bulk density $\rho$; (b) volumetric heat capacity during thawing $C_T$; (c) thermal conductivity of ground in thawed state $\lambda_t$; (d) thermal conductivity of ground in frozen state $\lambda_f$.

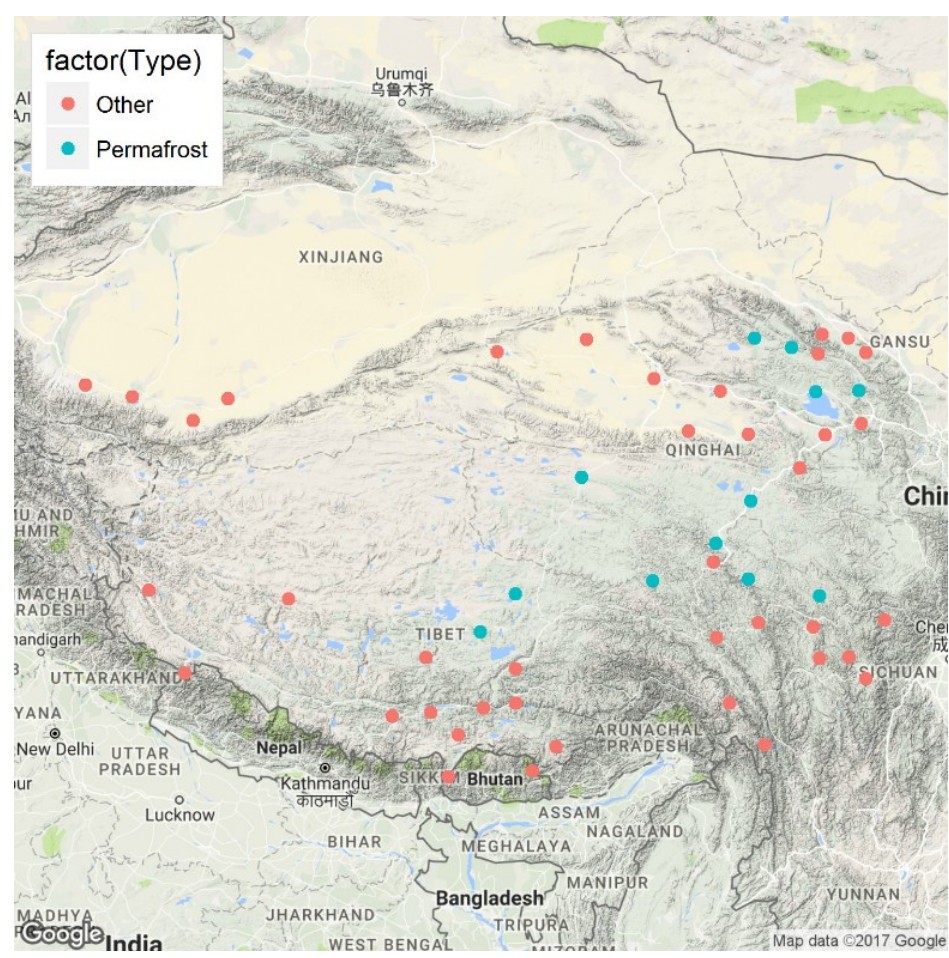

**Figure 4.** Permafrost occurrence map. Google Maps is a base map that uses the Exist_Permaforst function. "Other" indicates seasonal frozen soil.

**Table 4.** The statistical values of TTOP apply Com_Stats_QTP for the stations where permafrost exists. Intercept: y-intercept; Slope: slope of regression line; R: Pearson's correlation coefficient, $R^2$: coefficient of determination; RMSE: root mean squared error; NRMSE: normalized RMSE; SD_S: the standard deviation of TTOP using the Stefan function; SD_K: the standard deviation of TTOP using the Kudryavtsev function; MEF: modelling efficiency; NAE: normalized average error; VR: variance ratio; PBIAS: percent bias; NSE: Nash-Sutchliffe efficiency; RSR: RMSE-observations standard deviation ratio; and D: index of agreement.

| Statistic | Tuole | Wudaoliang | Anduo | Maduo | Qingshuihe | Shiqu |
|---|---|---|---|---|---|---|
| Intercept | -0.69 | -0.4 | -0.59 | -0.9 | -1.24 | -1.47 |
| Slope | 1.11 | 1.16 | 1.2 | 1.19 | 0.93 | 0.89 |
| R | 0.97 | 0.96 | 0.97 | 0.97 | 0.96 | 0.86 |
| $R^2$ | 0.94 | 0.92 | 0.93 | 0.94 | 0.92 | 0.75 |

| | | | | | |
|---|---|---|---|---|---|
| RMSE | 0.83 | 0.86 | 0.83 | 1.24 | 1.06 | 1.5 |
| NRMSE | -0.85 | -0.34 | -1.23 | -0.78 | -0.52 | -3.17 |
| SD_S | 0.59 | 0.8 | 0.78 | 0.61 | 1 | 0.69 |
| SD_K | 0.6 | 0.66 | 0.78 | 0.66 | 0.6 | 0.69 |
| MEF | -0.85 | 0.03 | -0.06 | -2.7 | 0.07 | -3.09 |
| NAE | 0.89 | 0.39 | 1.38 | 0.86 | 0.65 | 3.35 |
| VR | 1.03 | 0.68 | 1 | 1.14 | 0.35 | 1 |
| PBIAS | -76.13 | -26.54 | -108.59 | -67.31 | -41.42 | -255.56 |
| NSE | 0.42 | 0.62 | 0.57 | 0.39 | 0.67 | 0.37 |
| RSR | 0.76 | 0.61 | 0.66 | 0.78 | 0.58 | 0.79 |
| D | 0.67 | 0.7 | 0.76 | 0.53 | 0.58 | 0.5 |

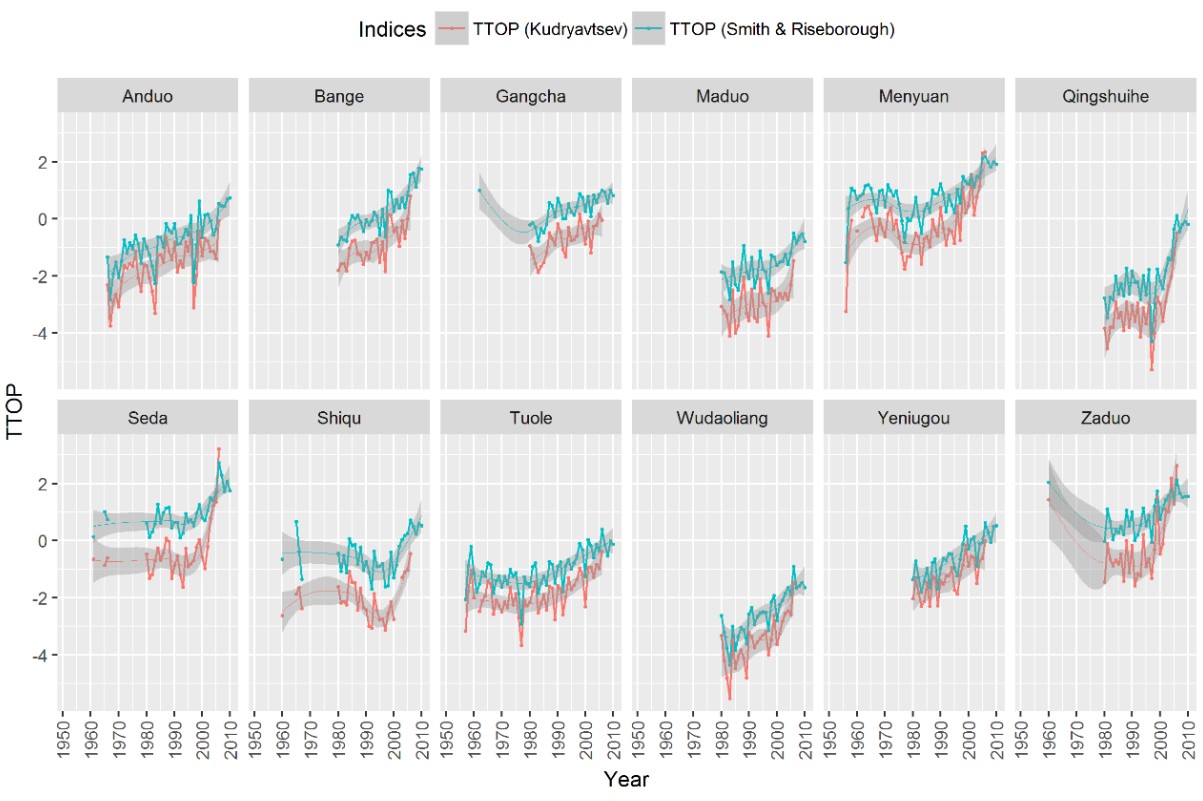

**Figure 5.** TTOP using the Smith and Kudryavtsev functions.

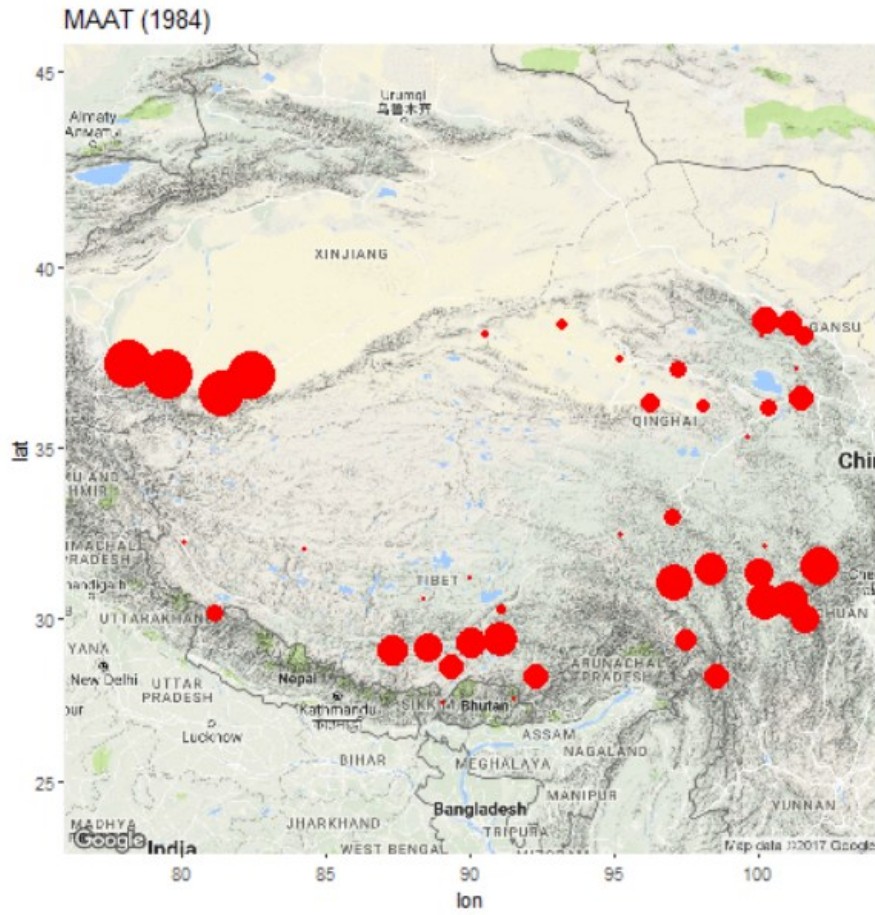

**Figure 6**. Index changes over time for MAAT. These graphs are animated in GIF mode.

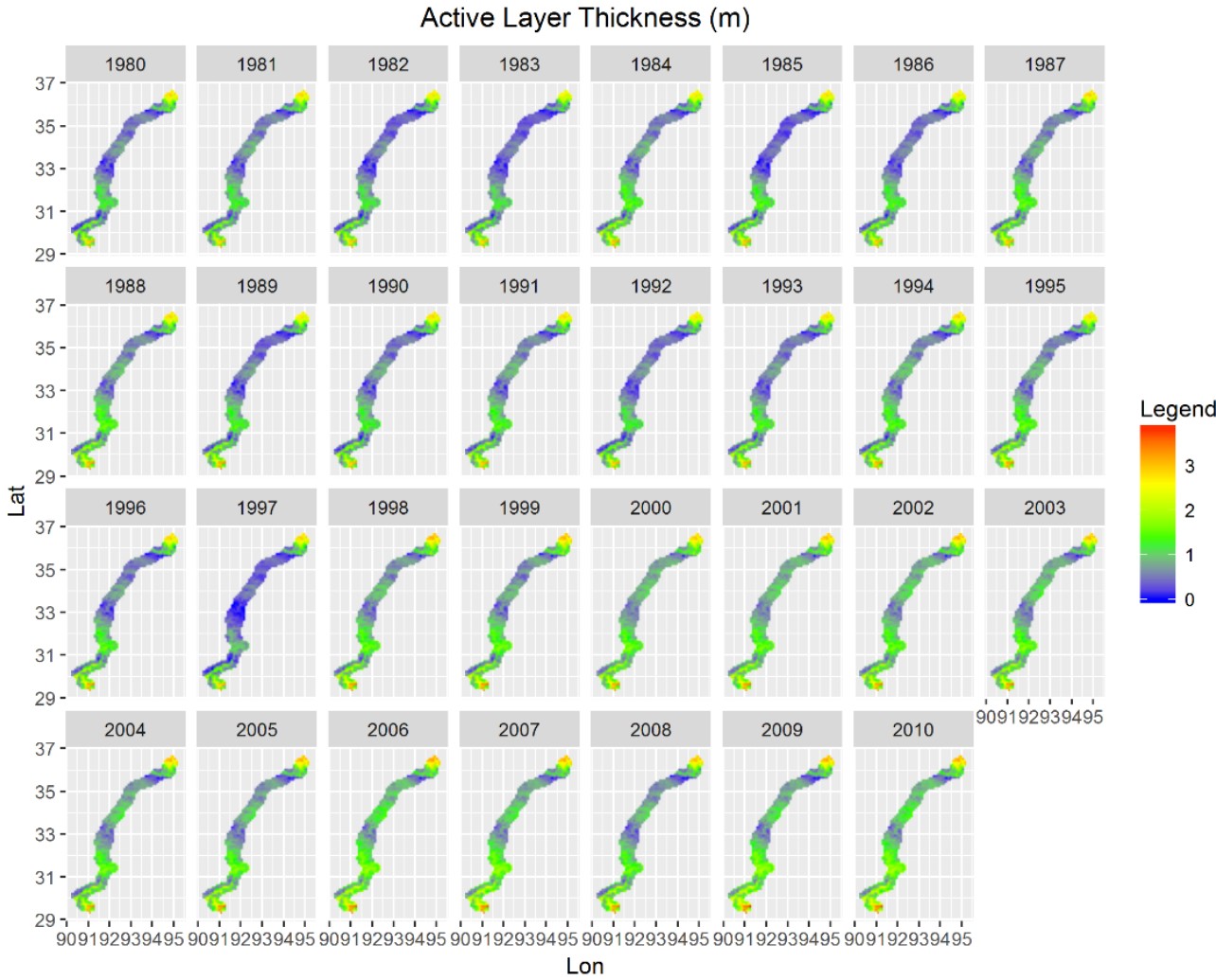

**Figure 7**. Regional visualization of ALT.

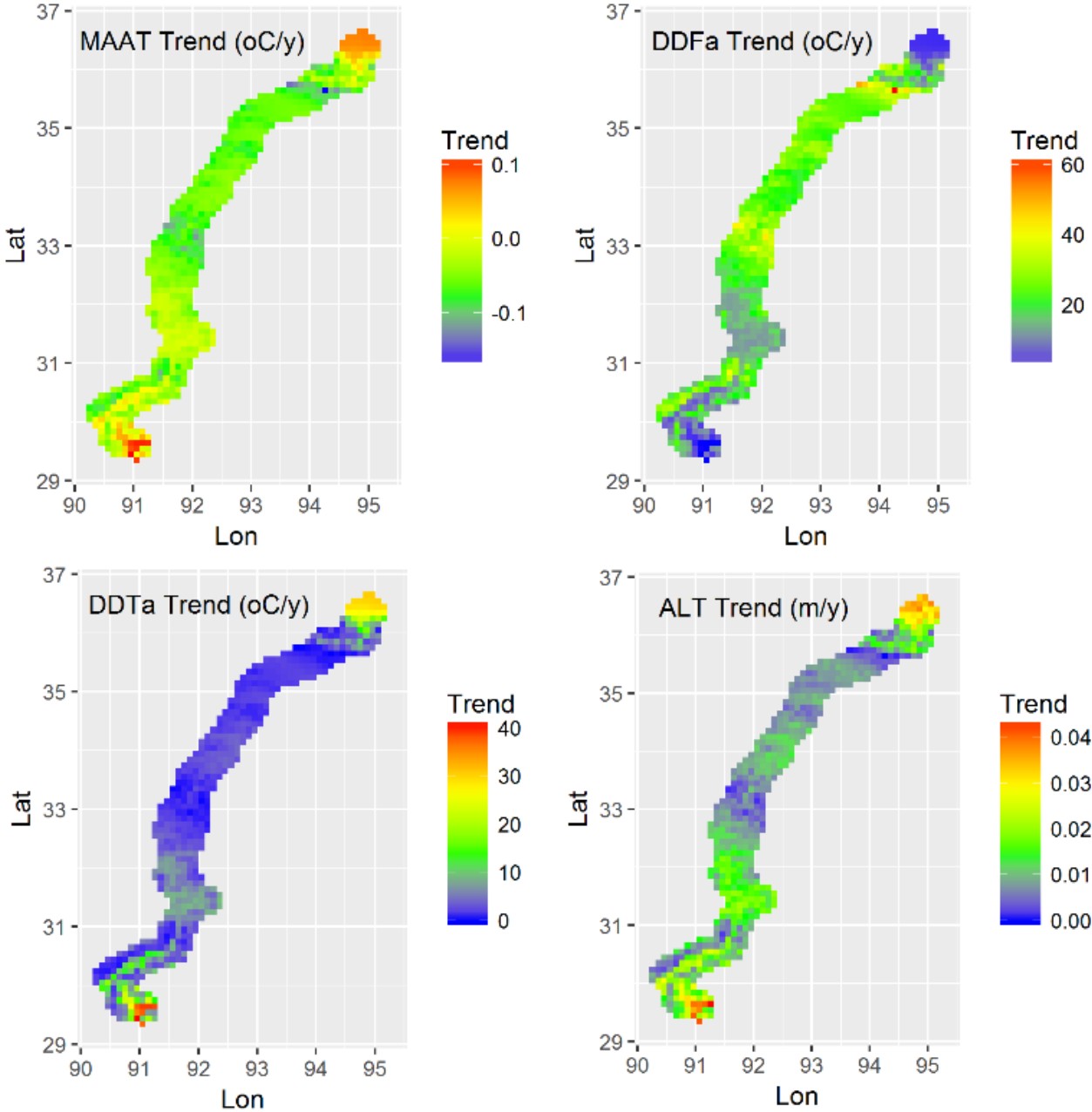

**Figure 8.** Spatial trend of MAAT, $DDT_a$, $DDF_a$, and ALT.

**Table 5**. The active layer thickness (ALT) and its trend between the PIC v1.3 simulation and literature analysis in the Qinghai-Tibet Engineering Corridor (QTEC).

| Mean | ALT | ALT trend | Period | Location | Data sources |
|------|-----|-----------|--------|----------|--------------|

| ALT (m) | Scope (m) | (cm yr$^{-1}$) | | | |
|---|---|---|---|---|---|
| 2.03 | 0.97-3.87 | 2.89 | 1980-2010 | The whole QTEC | PIC v1.3 |
| 2.18 | 1.00-3.20 | 1.33 | 1981-2010 | Near the Qinghai-Tibet highway along the QTEC | Li et al. (2012) |
| — | 1.00-3.00 | 0.50-2.00; 3.00-5.00 (1990s-2001) | 1980-2001 | Simulation along the Qinghai-Tibet Highway/ Railway | Oelke and Zhang (2007) |
| — | 1.30-3.50 | — | — | Near the Qinghai-Tibet highway along the QTEC | Pang et al. (2009) |
| — | 2.00-2.60 | 2.14-7.14 | 1991-1997 | 1 site (35°43′N, 94°05′E) Near the Qinghai-Tibet highway along the QTEC | Cheng and Wu (2007a) |
| — | 1.84-3.07 | — | 1990s | 17 Monitoring sites near the Qinghai-Tibet Highway/ Railway along the QTEC | Jin et al. (2008) |
| 2.41 | 1.32-4.57 | 7.50 | 1995-2007 | 10 Monitoring sites Near the Qinghai-Tibet highway along the QTEC | Wu and Zhang (2010) |
| 2.40 | 1.61-3.38 | 4.26 | 2002-2012 | 10 Monitoring sites (34°49′N, 92°55′E) along the QTEC | Wu et al. (2015) |