# Peer review of "PIC v1.3: Comprehensive R package for permafrost indices computing with daily weather observations and atmospheric forcing over the Qinghai-Tibet Plateau"

_Geoscientific Model Development, 2018_

## Short Comment (SC1) · 12 Mar 2018

1. Please include the version number for PIC in the title, and throughout the manuscript.

2. The precise version of the code discussed in the manuscript must be made available. The current best practice is for this code to be uploaded to a public repository and a DOI assigned. The DOI should be cited in the manuscript. github is inadequate because it does not readily link to the precise version of the code. However, making github code citable is not difficult; see: https://guides.github.com/activities/citable-code/

---

## Author Comment (AC1) · 13 Mar 2018

Thank you for the information related to code version and DOI. We will add the version number and organise a DOI for the revision of our manuscript. Thank you once again for your suggestions.

---

## Referee Comment (RC1) · Anonymous Referee #1 · 14 Apr 2018

General comments:

In this manuscript the authors introduce the "PIC" R-package for computing permafrost indices over the Qinghai-Tibet Plateau (QTP). The package can calculate 16 temperature/depth-related indices to estimate the possible change trends of frozen soil in the QTP, and provides over 10 statistical methods, a sequential Mann-Kendall trend test and spatial trend method to evaluate the permafrost indices. The package also provides multiple visual options to display the temporal and spatial variabilities on the stations and region. Along with the package, a dataset from 52 permanent meteorological stations across the QTP is prepared and the authors use it to demonstrate the temporal-spatial change trends of Tibetan permafrost with the climate.

The manuscript demonstrates some basic usages of PIC package. Although the authors state that the PIC package can be employed for a comprehensive analysis and can be used to validate the simulated results of the region, there's no such application presented in the current manuscript, therefore it's difficult to find the advantages of this package. On the other hand, a reasonable summarizing and categorizing the frozen indices developed in this package would be very useful for permafrost community, it's not available in the current manuscript. Overall, the manuscript is not well written and needs to be better organized. I do not recommend it for publication at the current stage, it could be reconsidered if the following points are addressed.

Specific comments:

1. It's mentioned that GLDAS and the weather station data of the surrounding QTEC were merged to produce a new data set, while it's not clear how this is done.

2. Please give concrete description on the parameters for the ground conditions, such as thermal conductivity of ground in thawed/frozen states, how were these parameters estimated or retrieved? their typical values and ranges at QTP.

3. It's mentioned several times that the PIC package integrates meteorological observations, remote sensing data, and field measurements to compute the factors or indices of permafrost and seasonal frozen soil. But from the manuscript, there's no description on how remote sensing data is integrated. It's also mentioned that the package integrates model simulations, it's not clear what model simulations refer to.

4. In Discussion, the authors state the simulation results from the PIC package show widespread permafrost degradation in QTP and the temporal-spatial trends of the permafrost conditions in QTP are consistent with previous studies. While there's no material results presented here to validate or compare with previous published literatures.

5. In Discussion, it's mentioned the spatial modeling at QTEC region classifies land cover and topographic features to determine the input spatial parameters, it's necessary to provide details and rationalities. It's also mentioned that the spatial modeling uses the GLDAS satellite data, but no detailed information.

6. The authors claim the PIC package will serve engineering applications and can be used to assess the impact of climate change on permafrost. Currently the package targets specifically QTP, how's the extensibility of this package? Is it possible to apply or extent the PIC package to other permafrost regions easily? If so, the PIC package will benefit a larger community.

Minor comments:

1. P2, L13: Change "Such an increase..." to "Such an increase in temperature of QTP..."

2. P2, L14-15: Add "Understanding" before "The distribution and changes of permafrost with climate...".

3. P3, L4: Change "depends on the size of" to "depends on the magnitude of"

4. P3, L15: Change "with" to "at".

5. P3, L16: Change "These indices consist..." to "The permafrost indices consist..."

6. P3, L19: Change "multi-dimensional simulation" to "multi-dimensional permafrost simulation"

7. P3, L21: Be more concise on the problem.

8. P3, L23: Change "the current condition" to "the current situation".

9. P3, L25: I doubt the word "determine" used here.

10. P5, L5: Change "function" to "functions".

11. P5, L6: Change "max and min" to "maximal and minimal".
12. P5, L14-20: The signs of equations from (3) to (6) are not consistent with equations (7) and (8).

13. P5, L21: Add "defined" before "in".

14. P8, L3: Please add a proper citation to R.

15. P8, L5: Change "functionality" to "functionalities".

16. P12, L9-10: What does "based on the 52 observation stations" mean? What index is used to detect the permafrost here?

17. P12, L11: Why does ALT decrease here?

18. P13, L8-11: It's better to mention that you're discussing technical implementation here. It will be more informative by giving the specification of the computer used to run the performance tests.

19. P13, L14-15: The point (2) is not clear.

20. P14, L1: Change "approximately" to "partially".

21. P14, L19: Please describe how the soil input parameters are handled in PIC directly.

22. Table 1: The units of thermal conductivity usually are written as "W m-1 K-1".

---

## Referee Comment (RC2) · Anonymous Referee #2 · 19 Apr 2018

Review Opinions GMD-2018-15, Luo L, et al. Comprehensive R Package for permafrost indices computations (PIC)

Overall problems English is problematic. Before resubmission, ask a native English speaker with good geoscience background to help edit the manuscript when all technical details are taken care of.

Specific issues: Title: OK Abstract: OK 1. Introduction P2, Lines 7-8, winter snow cover in some of those areas is supposed to one of the thickest in the world. P2,

Lines 14-16, sentence needs elaboration. The distribution and changes of permafrost with climate is necessary for infrastructure development, ecological and environmental assessments, and climate system modeling. The distribution of permafrost under influences of climate change is…. Notes: the epidemic issue here in the paper is rambunctious listing of references in the text. It should follow the GMD format, or at least the earlier, the first principle. Such as, Lines 10-11, 15-16, and others. Change them all and make the list more reasonable. P3, Lines 3-5, please cite original references, who proposed the classification of permafrost on the basis of MAGT in Chin and on the QTP? Additionally, it is on the MAGT, rather than on the size of the MAGT. What is the size of the MAGT? Page 3, Paragraph 15, The land surface temperature significantly differs the near-surface air temperatures and ground surface temperatures, particularly for the simulation of the thermal regime of ground. This is significant when taking into account of different driving input of the modeling. Please refer to Difference between near-surface air, land surface and ground surface temperatures and their influences on the frozen ground on the Qinghai-Tibet Plateau (Geoderma, Luo et al., 2018); Page 3, Line 20, please change "is a problem" to "problematic"; Page 5, Line 20, "MAGT is the soil temperature in (Wu and Zhang, 2010)." This sentence is incomplete.

Please also note the supplement to this comment:
https://www.geosci-model-dev-discuss.net/gmd-2018-15/gmd-2018-15-RC2-supplement.pdf

---

## Author Comment (AC2) · 7 May 2018

**"PIC v1.2: Comprehensive R package for permafrost indices computing with daily weather observations and atmospheric forcing over the Qinghai–Tibet Plateau"**

**by Lihui Luo et al.**

We thank Anonymous Reviewer #1 for the valuable feedback, which helped us to improve the manuscript. Please find below the Reviewer comments in black, Author responses in green, and Changes to the manuscript in blue.

**Response to reviewer comment 1:**

In this manuscript the authors introduce the "PIC" R-package for computing permafrost indices over the Qinghai-Tibet Plateau (QTP). The package can calculate temperature/depth-related indices to estimate the possible change trends of frozen soil in the QTP, and provides over 10 statistical methods, a sequential Mann-Kendall trend test and spatial trend method to evaluate the permafrost indices. The package also provides multiple visual options to display the temporal and spatial variabilities on the stations and region. Along with the package, a dataset from 52 permanent meteorological stations across the QTP is prepared and the authors use it to demonstrate the temporal-spatial change trends of Tibetan permafrost with the climate.

The manuscript demonstrates some basic usages of PIC package. Although the authors state that the PIC package can be employed for a comprehensive analysis and can be used to validate the simulated results of the region, there's no such application presented in the current manuscript, therefore it's difficult to find the advantages of this package. On the other hand, a reasonable summarizing and categorizing the frozen indices developed in this package would be very useful for permafrost community, it's not available in the current manuscript. Overall, the manuscript is not well written and needs to be better organized. I do not recommend it for publication at the current stage, it could be reconsidered if the following points are addressed.

Thanks for your insightful comments. In revising the paper, we have carefully considered your comments and suggestions. We agree with your comments on data, parameters, simulation verification, extensibility of the package, and so on. To address these concerns, we make the following modifications to the manuscript: (1) Reorganized the manuscript structure; (2) Added the preparation of datasets and parameters, and comparative analysis between simulations and observations; (3) Modified inappropriate expression; (4) Highlighted the importance of the transparency and repeatability in permafrost modeling, especially for the Qinghai-Tibet Plateau; (5) Improved the flow of the manuscript language (Figure R1). We tried our best to address each of your points in detail. We feel the revision represents an improvement and hope you do also. For more details, please see our replies below.

[Figure]

**EDITORIAL CERTIFICATE**

This document certifies that the manuscript listed below was edited for proper English language, grammar, punctuation, spelling, and overall style by one or more of the highly qualified native English speaking editors at American Journal Experts.

**Manuscript title:**
PIC v1.2: Comprehensive R package for permafrost indices computing with daily weather observations and atmospheric forcing over the Qinghai–Tibet Plateau

**Authors:**
Lihui Luo, Zhongqiong Zhang, Wei Ma, Shuhua Yi, Yanli Zhuang

**Date Issued:**
May 4, 2018

**Certificate Verification Key:**
0F39-05E1-2A03-E17C-F3E0

This certificate may be verified at www.aje.com/certificate. This document certifies that the manuscript listed above was edited for proper English language, grammar, punctuation, spelling, and overall style by one or more of the highly qualified native English speaking editors at American Journal Experts. Neither the research content nor the authors' intentions were altered in any way during the editing process. Documents receiving this certification should be English-ready for publication; however, the author has the ability to accept or reject our suggestions and changes. To verify the final AJE edited version, please visit our verification page. If you have any questions or concerns about this edited document, please contact American Journal Experts at support@aje.com.

American Journal Experts provides a range of editing, translation and manuscript services for researchers and publishers around the world. Our top-quality PhD editors are all native English speakers from America's top universities. Our editors come from nearly every research field and possess the highest qualifications to edit research manuscripts written by non-native English speakers. For more information about our company, services and partner discounts, please visit www.aje.com.

Figure R1. Editorial Certificate.

Specific comments:

1. It's mentioned that GLDAS and the weather station data of the surrounding QTEC were merged to produce a new data set, while it's not clear how this is done.

The main data processing workflow is as follows:

(1) Data pre-processing. There are a lot of details to be considered in the pre-processing workflow, such as time conversion, null value, unit conversion, and height correction in different datasets. For time conversion, China Meteorological Administration (CMA) data is based on Beijing time, while Beijing time is 8 hours earlier than time of Global Land Data Assimilation System (GLDAS). So the time of GLDAS data needs to be converted to coincide with CMA time. For height correction, the height of variables of the two datasets is different and needs to be revised according to the corresponding formula.

(2) Spatial interpolation of GLDAS. Higher spatial resolution data can be obtained through the spatial interpolation, which used bilinear interpolation method to implement spatial downscaling from 0.25° of GLDAS to 0.10°.

(3) Spatial interpolation of CMA. Spatial distribution of the CMA data can be obtained through transparent analysis and spatial interpolation of ground-based observations using ANUSPLIN package.

(4) Offset correction. The higher spatial resolution data was calibrated with correction parameters obtained from differences between the GLDAS data and the CMA data.

(5) Data post-processing. Post-processing mainly includes files segmentation, data compression, format conversion and so on.

We have updated the sentence as follows:

"The Qinghai-Tibet Engineering Corridor (QTEC), located at the centre of the QTP, was selected in preparing the atmospheric forcing data. Global Land Data Assimilation System (GLDAS, https://ldas.gsfc.nasa.gov) and the weather station data of the surrounding QTEC were merged through spatial interpolation and offset correction to produce a new data set for 1980 to 2010 with a daily 0.1° temporal-spatial resolution. An atmospheric forcing dataset was used as the input data for the PIC v1.2 regional calculation."

2. Please give concrete description on the parameters for the ground conditions, such as thermal conductivity of ground in thawed/frozen states, how were these parameters estimated or retrieved? their typical values and ranges at QTP.

We added the process of preparing the parameters in "3 Data and parameters" section.

"The parameters for the ground conditions were based on soil property data and field observations. The parameter data have two sets: one for weather stations and another for the QTEC region. The Harmonized World Soil Database (HWSD, version 1.21) provides information on soil parameters that are available for evaluating soil thermal conductivity with field observations and can be used as input parameters to the PIC v1.2 package (Bicheron et al., 2008; Nachtergaele et al., 2009). The thermal conductivity of ground in a thawed/frozen state, λt and λf, can be computed through the joint parameterization scheme of the Johansen method (Johansen, 1977) and Luo parameterization (Luo et al., 2009):

$$\lambda_{dry} = \frac{0.135 \times \rho + 64.7}{2700 - 0.947 \times \rho} \tag{25}$$

$$\lambda_s = \lambda_q{}^q \times \lambda_o{}^{1-q} \tag{26}$$

$$\lambda_{sat} = \lambda_s{}^{1-\theta_s} \times \lambda_w{}^{\theta_s} \tag{27}$$

$$S_r = \frac{\theta}{\theta_s} \tag{28}$$

$$K_{et} = \frac{K_t \times S_r}{1 + (K_t - 1) \times S_r} \tag{29}$$

$$K_{ef} = \frac{K_f \times S_r}{1 + (K_f - 1) \times S_r} \tag{30}$$

$$\lambda_t = (\lambda_{sat} - \lambda_{dry})K_{et} + \lambda_{dry} \tag{31}$$

$$\lambda_f = (\lambda_{sat} - \lambda_{dry})K_{ef} + \lambda_{dry} \tag{32}$$

where the soil thermal conductivity of dry soil $\lambda_{dry}$ depends on dry bulk density $\rho$, the thermal conductivity of soil solids $\lambda_s$ varies with the gravel content q, $\lambda_q$ is the thermal conductivity of quartz (7.7 W m$^{-1}$ K$^{-1}$), $\lambda_o$ is the thermal conductivity of other minerals (2.0 W m$^{-1}$ K$^{-1}$), and q is the gravel content in the soil. The saturated soil thermal conductivity $\lambda_{sat}$ depends on the thermal conductivity of soil solids $\lambda_s$, liquid water $\lambda_w$ (0.594 W m$^{-1}$ K$^{-1}$), and the soil saturated water content $\theta_s$. The degree of saturation $S_r$ is a function of the soil water content, $\theta$ and soil saturated water content, $\theta_s$. The Kersten numbers in the thawed/frozen state, $K_{et}$ and $K_{ef}$, are two functions of the degree of saturation $S_r$, and K values in the thawed/frozen state, $K_t$ and $K_f$; $\rho$, q and $\theta_s$ come from the T_BULK_DENSITY, T_GRAVEL, and T_BS fields of the HWSD. $\theta$, $K_t$ and $K_f$ in different soil textures can be found in Table 3. Figure 3 shows these parameters over the QTP.

Table 3: Parameters of thermal conductivity in the thawed/frozen state. The UADS Code came from soil

texture classification of United States Department of Agriculture (USDA). The Qinghai-Tibet Plateau does not have the 1 and 8 of soil classification codes. θ: soil water content; $K_t$: thermal conductivity of soil solid in thawed state; $K_f$: thermal conductivity of soil solid in frozen state.

| USDA Code | Soil Texture | $\theta$ | $K_t$ | $K_f$ |
|---|---|---|---|---|
| 1 | clay(heavy) | 0.17 | 1.90 | 0.85 |
| 2 | silty clay | 0.17 | 1.90 | 0.85 |
| 3 | clay (light) | 0.17 | 1.90 | 0.85 |
| 4 | silty clay loam | 0.17 | 1.90 | 0.85 |
| 5 | clay loam | 0.17 | 1.90 | 0.85 |
| 6 | silt | 0.17 | 1.90 | 0.85 |
| 7 | silt loam | 0.17 | 1.90 | 0.85 |
| 8 | sandy clay | 0.15 | 3.55 | 0.85 |
| 9 | loam | 0.15 | 3.55 | 0.95 |
| 10 | sandy clay loam | 0.15 | 3.55 | 0.95 |
| 11 | sandy loam | 0.15 | 3.55 | 0.95 |
| 12 | loamy sand | 0.06 | 4.60 | 1.70 |
| 13 | sand | 0.06 | 4.60 | 1.70 |

[Figure]

Figure 3: Spatial parameters for PIC v1.2 over the Qinghai-Tibet Plateau. (a) soil texture classification based on HWSD data; (b) dry bulk density ρ; (c) soil saturated water content $\theta$s; (d) thermal conductivity of dry soil λdry; (e) thermal conductivity of soil solids λs; (f) saturated soil thermal conductivity λsat; (g) thermal conductivity of ground in thawed state λt; (h) thermal conductivity of ground in frozen state λf."

In Discussion section "5.3 Limitations and uncertainties", we added parameter uncertainties.
"Second, the heterogeneity in ground conditions of the QTP also brings along uncertainties of parameter preparation."

3. It's mentioned several times that the PIC package integrates meteorological observations, remote sensing data, and field measurements to compute the factors or indices of permafrost and seasonal frozen soil. But from the manuscript, there's no description on how remote sensing data is integrated. It's also mentioned that the package integrates model simulations, it's not clear what model simulations refer to.

The Global Land Data Assimilation System (GLDAS) data and Harmonized World Soil Database (HWSD) came from remote sensing data. The spatial data with GLDAS and weather station data was called the gridded meteorological and soil datasets could be a more precise description. We changed "remote sensing data" to "gridded meteorological and soil datasets".
Model simulations, in fact, is the operation of the PIC package. We changed "model simulation" to "permafrost modeling" or "permafrost". In some statements, we still keep "simulation" the word.

4. In Discussion, the authors state the simulation results from the PIC package show widespread permafrost degradation in QTP and the temporal-spatial trends of the permafrost conditions in QTP are consistent with previous studies. While there's no material results presented here to validate or compare with previous published literatures.

The QTEC is the most accessible area of the QTP. Most boreholes were drilled in the QTEC to monitor changes of permafrost conditions, and this monitoring data provides support for model performance evaluation. Figure 7 and 8 provide the temporal-spatial change trends of the permafrost conditions using active layer thickness (ALT), and we added Table 5 to evaluated the PIC v1.2 simulation performance in "5.1 PIC performance"

"Climate change indicates a pronounced warming and permafrost degradation in the QTP with active layer deepening (Chen et al., 2013; Cheng and Wu, 2007b; Wu and Zhang, 2010; Wu et al., 2010), and both the simulation of stations and the region in PIC v1.2 also show widespread permafrost degradation (Figures 4-8). Meanwhile, as shown in Figures 7 & 8, the permafrost in the QTEC also continued to thaw, with the ALT growing. The QTEC is the most accessible area of the QTP. Most boreholes were drilled in the QTEC to monitor changes of permafrost conditions, and this monitoring data provides support for model performance evaluation. Meanwhile, ALT was widely used, so we adopted the permafrost index to estimate PIC v1.2 simulation performance. The simulated PIC v1.2 ALT and previous literature in the QTEC are compared in Table 5. The increasing rate of ALT averaged 0.50-7.50 cm yr$^{-1}$. The rate during the 1990s to 2010s was greater at more than 4.00 cm yr$^{-1}$, than during 1980 to the 1990s, at approximately 2.00 cm yr$^{-1}$. Though both the observed and the simulated ALT and its variation in different locations of the QTEC were still relatively large, the ALT trend in PIC v1.2 was close to the observations and simulation in the QTEC. In recent decades, the permafrost thaw rate has increased significantly. The majority of observed ALT and its trend along the QTH and QTR were greater than the simulated grid ALT of PIC v1.2, mainly because the observation sites are near these engineering facilities. These comparative analyses suggest that the temporal-spatial trends of permafrost conditions in the QTEC using PIC v1.2 were consistent with previous studies. More importantly, the difference between

simulation results highlights the importance of transparency and reusability of models, data, parameters, simulation results and so on."

Table 5. The active layer thickness (ALT) and its trend between the PIC v1.2 simulation and literature analysis in the Qinghai-Tibet Engineering Corridor (QTEC).

| Mean ALT (m) | ALT Scope (m) | ALT trend (cm yr-1) | Period | Location | Data sources |
|---|---|---|---|---|---|
| 2.03 | 0.97-3.87 | 2.89 | 1980-2010 | The whole QTEC | PIC v1.2 |
| 2.18 | 1.00-3.20 | 1.33 | 1981-2010 | Near the Qinghai-Tibet highway along the QTEC | Li et al. (2012) |
| — | 1.00-3.00 | 0.50-2.00; 3.00-5.00 (1990s-2001) | 1980-2001 | Simulation along the Qinghai-Tibet Highway/Railway | Oelke and Zhang (2007) |
| — | 1.30-3.50 | — | — | Near the Qinghai-Tibet highway along the QTEC | Pang et al. (2009) |
| — | 2.00-2.60 | 2.14-7.14 | 1991-1997 | 1 site (35°43'N, 94°05'E) Near the Qinghai-Tibet highway along the QTEC | Cheng and Wu (2007a) |
| — | 1.84-3.07 | — | 1990s | 17 Monitoring sites near the Qinghai-Tibet Highway/ Railway along the QTEC | Jin et al. (2008) |
| 2.41 | 1.32-4.57 | 7.50 | 1995-2007 | 10 Monitoring sites Near the Qinghai-Tibet highway along the QTEC | Wu and Zhang (2010) |
| 2.40 | 1.61-3.38 | 4.26 | 2002-2012 | 10 Monitoring sites (34°49'N, 92°55'E) along the QTEC | Wu et al. (2015) |

5. In Discussion, it's mentioned the spatial modeling at QTEC region classifies land cover and topographic features to determine the input spatial parameters, it's necessary to provide details and rationalities. It's also mentioned that the spatial modeling uses the GLDAS satellite data, but no detailed information.

Please see the our responses "1" and "2" in the Specific comments section.

6. The authors claim the PIC package will serve engineering applications and can be used to assess the impact of climate change on permafrost. Currently the package targets specifically QTP, how's the extensibility of this package? Is it possible to apply or extent the PIC package to other permafrost regions easily? If so, the PIC package will benefit a larger community.

The transparency and repeatability of data, parameters, model codes, computational processes, simulation output, visualization, and statistical analysis is a fundamental principle of scientific researches in the Earth system modeling. At present, there is a lack of open source software for permafrost modeling in the Tibetan plateau. The PIC v1.1 package use commonly used data and parameters, and these

permafrost indices are also applied to other permafrost regions; data and parameters of station calculation support a variety of data formats, while the current spatial data and parameters of region calculation only support NetCDF format, but this format is widely used in the Earth System Modeling community. A total of 52 weather stations with daily meteorological records over the Qinghai-Tibet Plateau (i.e., from 1951 to 2010) were integrated into the PIC package, which was never before. Whether it's data or packages, it can cause broad interest in permafrost communities. In order to ensure that the PIC package can be widely used, although the Qinghai-Tibet Plateau data is the default option, but are not confined to the Qinghai-Tibet Plateau, the invocation of PIC functions take into account the convenience of users. More importantly, these indices can be used separately to make free choices based on the needs of the researchers. Furthermore, we use the GNU-GPL 3.0 license, which other researchers can modify, refine, or integrate the PIC package into other software or Web service. Meanwhile, our team will continue to refine the package to meet a variety of needs.

Below, we will use an example to show the application of external data using PIC package, which came from other permafrost regions.

We used weather station data in northeastern China to compute permafrost indices using PIC package (please see Table R1 & Figure R2). We think that the PIC package can be fully extended to other areas.

Table R1. Station information for Mohe.

| SID | Station Name | Latitude | Longitude | Elevation | Start date | | End date | |
|-----|--------------|----------|-----------|-----------|------------|---|----------|---|
| 50136 | Mohe | 52°58' | 122°31' | 433 | 1958 | 1 | 2000 | 12 |

Figure R2. The computing process using PIC package. In this example, 1958 years of data quality is not good, because the missing value too much.

We added some sentences in Abstract and Discussion sections.

"The transparency and repeatability of the PIC v1.2 package and its data can be used and extended to assess the impact of climate change on permafrost."

"Moreover, the regional calculation can extend from QTEC to the entire QTP and even the other permafrost regions."

Minor comments:

1. P2, L13: Change "Such an increase. . ." to "Such an increase in temperature of QTP. . ."

This has been corrected, thank you.

2. P2, L14-15: Add "Understanding" before "The distribution and changes of permafrost with climate. . .".

This has been corrected, thank you.

3. P3, L4: Change "depends on the size of" to "depends on the magnitude of"

This has been corrected, thank you.

4. P3, L15: Change "with" to "at".

This has been corrected, thank you.

5. P3, L16: Change "These indices consist. . ." to "The permafrost indices consist. . ."

This has been corrected, thank you.

6. P3, L19: Change "multi-dimensional simulation" to "multi-dimensional permafrost simulation"

This has been corrected, thank you.

7. P3, L21: Be more concise on the problem.

We modified these sentence below.

"The transparency and repeatability of data, parameters, model codes, computational processes, simulation output, visualization, and statistical analysis is a fundamental principle of scientific research in Earth system modelling. At present, there is a lack of open source software, shared data and parameters for permafrost modelling in the QTP."

8. P3, L23: Change "the current condition" to "the current situation".

This has been corrected, thank you.

9. P3, L25: I doubt the word "determine" used here.

This has been corrected, thank you.
We have updated the sentence as follows:

"The goal is to provide guidance for the future of highway and high-speed railway design and construction in the QTP, as well as to further understand the effects of climate change on permafrost

dynamics."

10. P5, L5: Change "function" to "functions".

This has been corrected, thank you.

11. P5, L6: Change "max and min" to "maximal and minimal".

This has been corrected, thank you.

12. P5, L14-20: The signs of equations from (3) to (6) are not consistent with equations (7) and (8).

We checked these equations.

13. P5, L21: Add "defined" before "in".

This has been corrected, thank you. For a clearer description of MAGT, we rewrote the sentence as follows:
"MAGT is defined as the soil temperature at the depth of zero annual temperature change. $T_{z,t}$ is the ground temperature at any time $t$ and depth $z$ below a ground surface. MAGT is often found at the depths from 10 m to 16 m over the QTP (Wu and Zhang, 2010). Here, we take the $z$ value of 15 meters."

14. P8, L3: Please add a proper citation to R.

we have added the following as a reference to the manuscript:

R Core Team: R: A Language and Environment for Statistical Computing, R Foundation for Statistical Computing, Vienna, Austria, http: //www.R-project.org/, 2017.

15. P8, L5: Change "functionality" to "functionalities".

This has been corrected, thank you.

16. P12, L9-10: What does "based on the 52 observation stations" mean? What index is used to detect the permafrost here?

Using Exist_Permafrost function to detect the probability of permafrost occurrence. This has been corrected, thank you. We have updated the sentence as follows:

"The PIC v1.2 simulation results using the Exist_Permafrost function show that permafrost was detected at 12 of the 52 observation stations (Figure 4)"

17. P12, L11: Why does ALT decrease here?

Sorry. Wrong writing. Thanks for pointing this out. Have been modified to "increasing ALT".

"The permafrost, whether in permafrost stations or QTEC, continued to thaw with increasing ALT, low surface offset and thermal offset, and high MAAT, MAGST, MAGT, and TTOP for most areas of QTP."

18. P13, L8-11: It's better to mention that you're discussing technical implementation here. It will be more informative by giving the specification of the computer used to run the performance tests.

PIC v1.2 was run natively as a single process in Windows 7 Operating system. The calculations were performed independently through RStudio Desktop v1.1 software (RStudio, Inc., USA). The utilized processor type is Intel Core i7-2600 CPU 3.40GHz, and the available memory is 32 GB.

"The "for" loop is discarded, whereas the "apply" functions are used extensively to significantly lower the computation time. PIC v1.2 was run natively as a single process in the Windows 7 Operating system. The calculations were performed independently through RStudio Desktop v1.1 software (RStudio, Inc., USA). The utilized processor is an Intel Core i7-2600 CPU 3.40 GHz, and the available memory is 32 GB. The current regional calculation takes only approximately 11 s. Apart from the Kudryavtsev model that requires considerable computation time (i.e., approximately 5 min), the station calculation also exhibited an improved efficiency. Therefore, PIC v1.2 can be considered an efficient R package."

19. P13, L14-15: The point (2) is not clear.

MAGT is soil temperature at the depth of zero annual temperature change, which is often found at the depth from 10-16 m on the QTP. Regression analysis shows that MAGT on the QTP has the relationship as equation (R1-R3):

$$MAGT = -0.83\varphi - 0.0049E + 50.63341 \qquad (R1)$$
$$MAGT = 68.827 - 0.00827E - 0.927\varphi \qquad (R2)$$
$$MAGT = 65.461 - 1.222\varphi - 0.005E - 0.299\cos\theta \qquad (R3)$$

Where $\varphi$, E and $\theta$ represents latitude, elevation and aspect respectively.
We have updated the sentence as follows:

"(2) constructed a regression analysis method through the relationship between MAGT and elevation, latitude, and slope-aspects that presented a static permafrost distribution (Lu et al., 2013; Nan, 2005)."

20. P14, L1: Change "approximately" to "partially".

This has been corrected, thank you.

21. P14, L19: Please describe how the soil input parameters are handled in PIC directly.

Please see the our responses "2" in the Specific comments section (above).

22. Table 1: The units of thermal conductivity usually are written as "W m-1 K-1".

This has been corrected, thank you.

[revised manuscript text omitted]

---

## Author Comment (AC3) · 7 May 2018

please find our replies in supplement

Please also note the supplement to this comment:
https://www.geosci-model-dev-discuss.net/gmd-2018-15/gmd-2018-15-AC3-supplement.pdf

---

## Author Response (AR1)

**"PIC v1.3: Comprehensive R package for permafrost indices computing with daily weather observations and atmospheric forcing over the Qinghai–Tibet Plateau"**

**by Lihui Luo et al.**

We thank Anonymous Reviewer #1 for the valuable feedback, which helped us to improve the manuscript. Please find below the Reviewer comments in black, Author responses in green, and Changes to the manuscript in blue.

**Response to reviewer comment 1:**

In this manuscript the authors introduce the "PIC" R-package for computing permafrost indices over the Qinghai-Tibet Plateau (QTP). The package can calculate temperature/depth-related indices to estimate the possible change trends of frozen soil in the QTP, and provides over 10 statistical methods, a sequential Mann-Kendall trend test and spatial trend method to evaluate the permafrost indices. The package also provides multiple visual options to display the temporal and spatial variabilities on the stations and region. Along with the package, a dataset from 52 permanent meteorological stations across the QTP is prepared and the authors use it to demonstrate the temporal-spatial change trends of Tibetan permafrost with the climate.

The manuscript demonstrates some basic usages of PIC package. Although the authors state that the PIC package can be employed for a comprehensive analysis and can be used to validate the simulated results of the region, there's no such application presented in the current manuscript, therefore it's difficult to find the advantages of this package. On the other hand, a reasonable summarizing and categorizing the frozen indices developed in this package would be very useful for permafrost community, it's not available in the current manuscript. Overall, the manuscript is not well written and needs to be better organized. I do not recommend it for publication at the current stage, it could be reconsidered if the following points are addressed.

Thanks for your insightful comments. In revising the paper, we have carefully considered your comments and suggestions. We agree with your comments on data, parameters, simulation verification, extensibility of the package, and so on. To address these concerns, we make the following modifications to the manuscript: (1) Reorganized the manuscript structure; (2) Added the preparation of datasets and parameters, and comparative analysis between simulations and observations; (3) Modified many inappropriate expression; (4) Highlighted the importance of the transparency and repeatability in permafrost modeling, especially for the current permafrost study in the Qinghai-Tibet Plateau, openness and sharing are extremely important from data, parameters, model codes, computational processes, simulation output, statistical analysis to visualization; (5) Improved the flow of the manuscript language (Figure R1). We tried our best to address each of your points in detail. We feel the revision represents an

improvement and hope you do also. For more details, please see our replies below.

[Figure]

[Figure]

EDITORIAL CERTIFICATE

This document certifies that the manuscript listed below was edited for proper English language, grammar, punctuation, spelling, and overall style by one or more of the highly qualified native English speaking editors at American Journal Experts.

Manuscript title:
PIC v1.2: Comprehensive R package for permafrost indices computing with daily weather observations and atmospheric forcing over the Qinghai–Tibet Plateau

Authors:
Lihui Luo, Zhongqiong Zhang, Wei Ma, Shuhua Yi, Yanli Zhuang

Date Issued:
May 4, 2018

Certificate Verification Key:
0F39-05E1-2A03-E17C-F3E0

This certificate may be verified at www.aje.com/certificate. This document certifies that the manuscript listed above was edited for proper English language, grammar, punctuation, spelling, and overall style by one or more of the highly qualified native English speaking editors at American Journal Experts. Neither the research content nor the authors' intentions were altered in any way during the editing process. Documents receiving this certification should be English-ready for publication; however, the author has the ability to accept or reject our suggestions and changes. To verify the final AJE edited version, please visit our verification page. If you have any questions or concerns about this edited document, please contact American Journal Experts at support@aje.com.

American Journal Experts provides a range of editing, translation and manuscript services for researchers and publishers around the world. Our top-quality PhD editors are all native English speakers from America's top universities. Our editors come from nearly every research field and possess the highest qualifications to edit research manuscripts written by non-native English speakers. For more information about our company, services and partner discounts, please visit www.aje.com.

Figure R1. Editorial Certificate.

5    Specific comments:

1. It's mentioned that GLDAS and the weather station data of the surrounding QTEC were merged to produce a new data set, while it's not clear how this is done.

The main data processing workflow is as follows:

10    (1) Data pre-processing. There are a lot of details to be considered in the pre-processing workflow, such as time conversion, null value, unit conversion, and height correction in different datasets. For time conversion, China Meteorological Administration (CMA) data is based on Beijing time, while Beijing time is 8 hours earlier than time of Global Land Data Assimilation System (GLDAS). So the time of GLDAS data needs to be converted to coincide with CMA time. For height correction, the height of

15    variables of the two datasets is different and needs to be revised according to the corresponding formula.

(2) Spatial interpolation of GLDAS. Higher spatial resolution data can be obtained through the spatial interpolation, which used bilinear interpolation method to implement spatial downscaling from 0.25° of GLDAS to 0.10°.

(3) Spatial interpolation of CMA. Spatial distribution of the CMA data can be obtained through

20    transparent analysis and spatial interpolation of ground-based observations using ANUSPLIN package.

(4) Offset correction. The higher spatial resolution data was calibrated with correction parameters obtained from differences between the GLDAS data and the CMA data.

(5) Data post-processing. Post-processing mainly includes files segmentation, data compression, format conversion and so on.

We have updated the sentence as follows:

"The Qinghai-Tibet Engineering Corridor (QTEC), located at the centre of the QTP, was selected in preparing the atmospheric forcing data. Global Land Data Assimilation System (GLDAS, https://ldas.gsfc.nasa.gov) and the weather station data of the surrounding QTEC were merged through spatial interpolation and offset correction to produce a new data set for 1980 to 2010 with a daily 0.1° temporal-spatial resolution. An atmospheric forcing dataset was used as the input data for the PIC v1.3 regional calculation."

2. Please give concrete description on the parameters for the ground conditions, such as thermal conductivity of ground in thawed/frozen states, how were these parameters estimated or retrieved? their typical values and ranges at QTP.

We added the process of preparing the parameters in "3 Data and parameters" section. The computing parameters for whole processing can be found in Figure R2. We selected 4 input spatial parameters as Figure 3 of the manuscript.

[Figure]

Figure R2: Computing parameters for PIC v1.3 over the Qinghai-Tibet Plateau. (a) soil texture classification based on HWSD data; (b) dry bulk density ρ; (c) soil saturated water content $\theta$s; (d) thermal conductivity of dry soil λ_dry; (e) thermal conductivity of soil solids λ_s; (f) saturated soil thermal

conductivity $\lambda_{sat}$; (g) thermal conductivity of ground in thawed state $\lambda_t$; (h) thermal conductivity of ground in frozen state $\lambda_f$; (i) volumetric heat capacity during thawing $C_T$.

**3.3 Parameters**

[revised manuscript text omitted]

volumetric heat capacity during thawing CT; (c) thermal conductivity of ground in thawed state λt; (d) thermal conductivity of ground in frozen state λf.

In Discussion section "5.3 Limitations and uncertainties", we added parameter uncertainties.
"Second, the heterogeneity in ground conditions of the QTP also brings along uncertainties of parameter preparation."

3. It's mentioned several times that the PIC package integrates meteorological observations, remote sensing data, and field measurements to compute the factors or indices of permafrost and seasonal frozen soil. But from the manuscript, there's no description on how remote sensing data is integrated. It's also mentioned that the package integrates model simulations, it's not clear what model simulations refer to.

The Global Land Data Assimilation System (GLDAS) data and Harmonized World Soil Database (HWSD) came from remote sensing data. The spatial data with GLDAS and weather station data was called the gridded meteorological and soil datasets could be a more precise description. We changed "remote sensing data" to "gridded meteorological and soil datasets".
Model simulations, in fact, is the operation of the PIC package. We changed "model simulation" to "permafrost modeling". In some statements, we still keep "simulation" the word.

4. In Discussion, the authors state the simulation results from the PIC package show widespread permafrost degradation in QTP and the temporal-spatial trends of the permafrost conditions in QTP are consistent with previous studies. While there's no material results presented here to validate or compare with previous published literatures.

The QTEC is the most accessible area of the QTP. Most boreholes were drilled in the QTEC to monitor changes of permafrost conditions, and this monitoring data provides support for model performance evaluation. Figure 7 and 8 provide the temporal-spatial change trends of the permafrost conditions using active layer thickness (ALT), and we added Table 5 to evaluated the PIC v1.3 simulation performance in "5.1 PIC performance"

"Climate change indicates a pronounced warming and permafrost degradation in the QTP with active layer deepening (Chen et al., 2013; Cheng and Wu, 2007b; Wu and Zhang, 2010; Wu et al., 2010), and both the simulation of stations and the region in PIC v1.3 also show widespread permafrost degradation (Figures 4-8). Meanwhile, as shown in Figures 7 & 8, the permafrost in the QTEC also continued to thaw, with the ALT growing. The QTEC is the most accessible area of the QTP. Most boreholes were drilled in the QTEC to monitor changes of permafrost conditions, and this monitoring data provides support for model performance evaluation. Meanwhile, ALT was widely used, so we adopted the permafrost index to estimate PIC v1.3 simulation performance. The simulated PIC v1.3 ALT and previous literature in the QTEC are compared in Table 5. The increasing rate of ALT averaged 0.50-7.50 cm yr$^{-1}$. The rate during the 1990s to 2010s was greater at more than 4.00 cm yr$^{-1}$, than during 1980 to the 1990s, at approximately 2.00 cm yr$^{-1}$. Though both the observed and the simulated ALT and its variation in different locations of the QTEC were still relatively large, the ALT trend in PIC v1.3 was close to the observations and simulation in the QTEC. In recent decades, the permafrost thaw rate has increased significantly. The majority of observed ALT and its trend along the QTH and QTR were greater than the simulated grid

ALT of PIC v1.3, mainly because the observation sites are near these engineering facilities. These comparative analyses suggest that the temporal-spatial trends of permafrost conditions in the QTEC using PIC v1.3 were consistent with previous studies. More importantly, the difference between simulation results highlights the importance of transparency and reusability of models, data, parameters, simulation results and so on."

Table 5. The active layer thickness (ALT) and its trend between the PIC v1.3 simulation and literature analysis in the Qinghai-Tibet Engineering Corridor (QTEC).

| Mean ALT (m) | ALT Scope (m) | ALT trend (cm yr-1) | Period | Location | Data sources |
|---|---|---|---|---|---|
| 2.03 | 0.97-3.87 | 2.89 | 1980-2010 | The whole QTEC | PIC v1.3 |
| 2.18 | 1.00-3.20 | 1.33 | 1981-2010 | Near the Qinghai-Tibet highway along the QTEC | Li et al. (2012) |
| — | 1.00-3.00 | 0.50-2.00; 3.00-5.00 (1990s-2001) | 1980-2001 | Simulation along the Qinghai-Tibet Highway/Railway | Oelke and Zhang (2007) |
| — | 1.30-3.50 | — | — | Near the Qinghai-Tibet highway along the QTEC | Pang et al. (2009) |
| — | 2.00-2.60 | 2.14-7.14 | 1991-1997 | 1 site (35°43'N, 94°05'E) Near the Qinghai-Tibet highway along the QTEC | Cheng and Wu (2007a) |
| — | 1.84-3.07 | — | 1990s | 17 Monitoring sites near the Qinghai-Tibet Highway/ Railway along the QTEC | Jin et al. (2008) |
| 2.41 | 1.32-4.57 | 7.50 | 1995-2007 | 10 Monitoring sites Near the Qinghai-Tibet highway along the QTEC | Wu and Zhang (2010) |
| 2.40 | 1.61-3.38 | 4.26 | 2002-2012 | 10 Monitoring sites (34°49'N, 92°55'E) along the QTEC | Wu et al. (2015) |

5. In Discussion, it's mentioned the spatial modeling at QTEC region classifies land cover and topographic features to determine the input spatial parameters, it's necessary to provide details and rationalities. It's also mentioned that the spatial modeling uses the GLDAS satellite data, but no detailed information.

Please see the our responses "1" ,"2" and "3" in the Specific comments section.

6. The authors claim the PIC package will serve engineering applications and can be used to assess the impact of climate change on permafrost. Currently the package targets specifically QTP, how's the extensibility of this package? Is it possible to apply or extent the PIC package to other permafrost regions easily? If so, the PIC package will benefit a larger community.

The transparency and repeatability of data, parameters, model codes, computational processes,

simulation output, visualization, and statistical analysis is a fundamental principle of scientific researches in the Earth system modeling. At present, there is a lack of open source software for permafrost modeling in the Tibetan plateau. The PIC v1.1 package use commonly used data and parameters, and these permafrost indices are also applied to other permafrost regions; data and parameters of station calculation

5 support a variety of data formats, while the current spatial data and parameters of region calculation only support NetCDF format, but this format is widely used in the Earth System Modeling community. A total of 52 weather stations with daily meteorological records over the Qinghai-Tibet Plateau (i.e., from 1951 to 2010) were integrated into the PIC package, which was never before. Whether it's data or packages, it can cause broad interest in permafrost communities. In order to ensure that the PIC package can be widely

10 used, although the Qinghai-Tibet Plateau data is the default option, but are not confined to the Qinghai-Tibet Plateau, the invocation of PIC functions take into account the convenience of users. More importantly, these indices can be used separately to make free choices based on the needs of the researchers. Furthermore, we use the GNU-GPL 3.0 license, which other researchers can modify, refine, or integrate the PIC package into other software or Web service. Meanwhile, our team will continue to

15 refine the package to meet a variety of needs.

Below, we will use an example to show the application of external data using PIC package, which came from other permafrost regions.

We used weather station data in northeastern China to compute permafrost indices using PIC package (please see Table R1, Figure R3 & R4). We think that the PIC package can be fully extended to other

20 areas.

Table R1. Station information for Mohe station.

| SID | Station Name | Latitude | Longitude | Elevation | Start date | | End date | |
|-----|--------------|----------|-----------|-----------|------------|---|----------|---|
| 50136 | Mohe | 52°58' | 122°31' | 433 | 1958 | 1 | 2000 | 12 |

25 Figure R3. The computing process using PIC package for Mohe station. The other permafrost indices can also be computed.

[Figure]

Figure R4. The visualization example of the permafrost indices (MAAT, MAGST and MAGT) using PIC package for Mohe station.

We added some sentences in Abstract and Discussion sections.

"The transparency and repeatability of the PIC v1.3 package and its data can be used and extended to assess the impact of climate change on permafrost."

"Moreover, the regional calculation can extend from QTEC to the entire QTP and even the other permafrost regions."

Minor comments:

1. P2, L13: Change "Such an increase. . ." to "Such an increase in temperature of QTP. . ."

This has been corrected, thank you.

2. P2, L14-15: Add "Understanding" before "The distribution and changes of permafrost with climate. . .".

This has been corrected, thank you.

3. P3, L4: Change "depends on the size of" to "depends on the magnitude of"

This has been corrected, thank you.

4. P3, L15: Change "with" to "at".

This has been corrected, thank you.

5. P3, L16: Change "These indices consist. . ." to "The permafrost indices consist. . ."

This has been corrected, thank you.

6. P3, L19: Change "multi-dimensional simulation" to "multi-dimensional permafrost simulation"

This has been corrected, thank you.

7. P3, L21: Be more concise on the problem.

We modified these sentence below.

"The transparency and repeatability of data, parameters, model codes, computational processes, simulation output, visualization, and statistical analysis is a fundamental principle of scientific research in Earth system modelling. At present, there is a lack of open source software, shared data and parameters for permafrost modelling in the QTP."

8. P3, L23: Change "the current condition" to "the current situation".

This has been corrected, thank you.

9. P3, L25: I doubt the word "determine" used here.

This has been corrected, thank you.
We have updated the sentence as follows:

"The goal is to provide guidance for the future of highway and high-speed railway design and construction in the QTP, as well as to further understand the effects of climate change on permafrost dynamics."

10. P5, L5: Change "function" to "functions".

This has been corrected, thank you.

11. P5, L6: Change "max and min" to "maximal and minimal".

This has been corrected, thank you.

12. P5, L14-20: The signs of equations from (3) to (6) are not consistent with equations (7) and (8).

We checked these equations.

13. P5, L21: Add "defined" before "in".

This has been corrected, thank you. For a clearer description of MAGT, we rewrote the sentence as follows:
"MAGT is defined as the soil temperature at the depth of zero annual temperature change. $T_{z,t}$ is the ground temperature at any time $t$ and depth $z$ below a ground surface. MAGT is often found at the depths from 10 m to 16 m over the QTP (Wu and Zhang, 2010). Here, we take the z value of 15 meters as default

value, user can change the depth z."

14. P8, L3: Please add a proper citation to R.

we have added the following as a reference to the manuscript:

R Core Team: R: A Language and Environment for Statistical Computing, R Foundation for Statistical Computing, Vienna, Austria, http: //www.R-project.org/, 2017.

15. P8, L5: Change "functionality" to "functionalities".

This has been corrected, thank you.

16. P12, L9-10: What does "based on the 52 observation stations" mean? What index is used to detect the permafrost here?

Using Exist_Permafrost function to detect the probability of permafrost occurrence. This has been corrected, thank you. We have updated the sentence as follows:

"The PIC v1.3 simulation results using the Exist_Permafrost function show that permafrost was detected at 12 of the 52 observation stations (Figure 4)"

17. P12, L11: Why does ALT decrease here?

Sorry. Wrong writing. Thanks for pointing this out. Have been modified to "increasing ALT".

"The permafrost, whether in permafrost stations or QTEC, continued to thaw with increasing ALT, low surface offset and thermal offset, and high MAAT, MAGST, MAGT, and TTOP for most areas of QTP."

18. P13, L8-11: It's better to mention that you're discussing technical implementation here. It will be more informative by giving the specification of the computer used to run the performance tests.

PIC v1.3 was run natively as a single process in Windows 7 Operating system. The calculations were performed independently through RStudio Desktop v1.1 software (RStudio, Inc., USA). The utilized processor type is Intel Core i7-2600 CPU 3.40GHz, and the available memory is 32 GB.

"The "for" loop is discarded, whereas the "apply" functions are used extensively to significantly lower the computation time. PIC v1.3 was run natively as a single process in the Windows 7 Operating system. The calculations were performed independently through RStudio Desktop v1.1 software (RStudio, Inc., USA). The utilized processor is an Intel Core i7-2600 CPU 3.40 GHz, and the available memory is 32 GB. The current regional calculation takes only approximately 11 s. Apart from the Kudryavtsev model that requires considerable computation time (i.e., approximately 5 min), the station calculation also exhibited an improved efficiency. Therefore, PIC v1.3 can be considered an efficient R package."

19. P13, L14-15: The point (2) is not clear.

MAGT is soil temperature at the depth of zero annual temperature change, which is often found at 10–15 m depth below the ground surface on the QTP. Regression analysis shows that MAGT on the QTP has the relationship as equation (R1-R3):

$$MAGT = -0.83\varphi - 0.0049E + 50.63341 \tag{R1}$$
$$MAGT = 68.827 - 0.00827E - 0.927\varphi \tag{R2}$$
$$MAGT = 65.461 - 1.222\varphi - 0.005E - 0.299\cos\theta \tag{R3}$$

Where $\varphi$, E and $\theta$ represents latitude, elevation and aspect respectively.

We have updated the sentence as follows:

"(2) constructed a regression analysis method through the relationship between MAGT and elevation, latitude, and slope-aspects that presented a static permafrost distribution (Lu et al., 2013; Nan, 2005)."

20. P14, L1: Change "approximately" to "partially".

This has been corrected, thank you.

21. P14, L19: Please describe how the soil input parameters are handled in PIC directly.

Please see the our responses "2" in the Specific comments section (above).

22. Table 1: The units of thermal conductivity usually are written as "W m-1 K-1".

This has been corrected, thank you.

We thank Anonymous Reviewer #2 for the valuable feedback, which helped to improve the manuscript. Please find below the Reviewer comments in black, Author responses in green, and Changes to the manuscript in blue.

**Response to reviewer comment 2:**

Overall problems English is problematic. Before resubmission, ask a native English speaker with good geoscience background to help edit the manuscript when all technical details are taken care of.

Thanks for your insightful comments. In revising the paper, we have carefully considered your comments and suggestions. We agree with your comments on some details and the latest progress of permafrost modeling. To address these concerns, we make the following modifications to the manuscript: (1) Reorganized the manuscript structure; (2) Added the preparation of datasets and parameters, and comparative analysis between simulations and observations; (3) Modified inappropriate expression; (4) Highlighted the importance of the transparency and repeatability in permafrost modeling, especially for the current permafrost study in the Qinghai-Tibet Plateau, openness and sharing are extremely important from data, parameters, model codes, computational processes, simulation output, statistical analysis to visualization; (5) Improved the flow of the manuscript language (Figure R1). We tried our best to address each of your points in detail. We feel the revision represents an improvement and hope you do also. For more details, please see our replies below.

[Figure]

**EDITORIAL CERTIFICATE**

This document certifies that the manuscript listed below was edited for proper English language, grammar, punctuation, spelling, and overall style by one or more of the highly qualified native English speaking editors at American Journal Experts.

**Manuscript title:**
PIC v1.2: Comprehensive R package for permafrost indices computing with daily weather observations and atmospheric forcing over the Qinghai–Tibet Plateau

**Authors:**
Lihui Luo, Zhongqiong Zhang, Wei Ma, Shuhua Yi, Yanli Zhuang

**Date Issued:**
May 4, 2018

**Certificate Verification Key:**
0F39-05E1-2A03-E17C-F3E0

[Figure]

This certificate may be verified at www.aje.com/certificate. This document certifies that the manuscript listed above was edited for proper English language, grammar, punctuation, spelling, and overall style by one or more of the highly qualified native English speaking editors at American Journal Experts. Neither the research content nor the authors' intentions were altered in any way during the editing process. Documents receiving this certification should be English-ready for publication; however, the author has the ability to accept or reject our suggestions and changes. To verify the final AJE edited version, please visit our verification page. If you have any questions or concerns about this edited document, please contact American Journal Experts at support@aje.com.

American Journal Experts provides a range of editing, translation and manuscript services for researchers and publishers around the world. Our top-quality PhD editors are all native English speakers from America's top universities. Our editors come from nearly every research field and possess the highest qualifications to edit research manuscripts written by non-native English speakers. For more information about our company, services and partner discounts, please visit www.aje.com.

Figure R1. Editorial Certificate.

Specific issues: Title: OK Abstract: OK

1. Introduction P2, Lines 7-8, winter snow cover in some of those areas is supposed to one of the thickest in the world.

Thanks for pointing this out.

10    We have updated the sentence as follows:

"Permafrost occurs mostly in high latitudes and altitudes with long, cold winters and thick winter snow, e.g., the Arctic, Antarctica, Alaska, the Alps, Northern Russia, Northern Canada, Northern Mongolia, and the Qinghai-Tibet Plateau (QTP) (Riseborough et al., 2008; Yi et al., 2014; Zhang et al., 2008)"

P2, Lines 14-16, sentence needs elaboration. The distribution and changes of permafrost with climate is necessary for infrastructure development, ecological and environmental assessments, and climate system modeling. The distribution of permafrost under influences of climate change is. . .. Notes: the epidemic issue here in the paper is rambunctious listing of references in the text. It should follow the GMD format,

20    or at least the earlier, the first principle. Such as, Lines 10-11, 15-16, and others. Change them all and make the list more reasonable.

Thanks for pointing this out. We check the reference format.
We have updated the sentence as follows:

"Understanding the distribution and changes of permafrost under the influence of climate change is necessary for infrastructure development, ecological and environmental assessment, and climate system

modelling (Luo et al., 2017; Luo et al., 2012; Zhang et al., 2014)"

P3, Lines 3-5, please cite original references, who proposed the classification of permafrost on the basis of MAGT in Chin and on the QTP? Additionally, it is on the MAGT, rather than on the size of the MAGT. What is the size of the MAGT?

We add a reference. MAGT is often found at the depth from 10 m to 16 m over the QTP (Wu and Zhang, 2010), here we take the value of 15 meters. Usually, size connotes physical dimensions while magnitude connotes either a numerical measure of any sort of amount or metaphorical size. Our use of the word "size" is wrong. So we changed the "size" to "range".

"Thereafter, the type and distribution of frozen soil can be classified in a variety of manners depending on the range and magnitude of these indices."

Page 3, Paragraph 15, The land surface temperature significantly differs the near-surface air temperatures and ground surface temperatures, particularly for the simulation of the thermal regime of ground. This is significant when taking into account of different driving input of the modeling. Please refer to Difference between near-surface air, land surface and ground surface temperatures and their influences on the frozen ground on the Qinghai-Tibet Plateau (Geoderma, Luo et al., 2018);

Thanks for pointing this out. We agree with your comments on the difference of three temperature values, input data with different temperatures will cause the difference of simulation.
The Land Surface Temperature (LST) is the radiative skin temperature of the land surface, as measured in the direction of the remote sensor. LST is a mixture of vegetation and bare soil temperatures. The ground surface temperature (GST) is the soil temperature at 0–5 cm. The near-surface air temperature ($T_a$) was measured at a screen-height of 1.5–2 m.
In the PIC v 1.1 package, we use near-surface air temperature and ground surface temperature at 0 cm, which came from weather stations and GLDAS gridded meteorological datasets. In the future we will use spatial data of land/ground surface temperature as a input data of PIC package, and we will consider the simulation difference between LST and GST.

Page 3, Line 20, please change "is a problem" to "problematic";

We rewrote the sentence.

"The transparency and repeatability of data, parameters, model codes, computational processes, simulation output, visualization, and statistical analysis is a fundamental principle of scientific research in Earth system modelling. At present, there is a lack of open source software, shared data and parameters for permafrost modelling in the QTP."

Page 5, Line 20, "MAGT is the soil temperature in (Wu and Zhang, 2010)." This sentence is incomplete.

Thanks for pointing this out. For a clearer description of MAGT, we rewrote the sentence as follows:

"MAGT is defined as the soil temperature at the depth of zero annual temperature change. $T_{z,t}$ is the ground temperature at any time $t$ and depth $z$ below a ground surface. MAGT is often found at the depths from 10 m to 16 m over the QTP (Wu and Zhang, 2010). Here, we take the $z$ value of 15 meters as default value, user can change the depth $z$."

Please also note the supplement to this comment: https://www.geosci-model-dev-discuss.net/gmd-2018-15/gmd-2018-15-RC2- supplement.pdf

We moved the reviewer's comments here from the manuscript edits.

Page 2, Line 3, consecutive

This has been added, thank you.

15    Page 2, Line 7, permafrost occurs also in Alps where there is a considerable snow cover during the winter.

This has been added, thank you.

Page 2, Line 9, Need to check if for the inside cite, the journal request that multiple authors to be arranged

20    alphabetically and not by year.

Thanks. We check the reference format for the inside cite.

Page 2, Line 10, There are some other opinions in a recent paper. At least is adecvate to cite them. Ran

25    et al. 2018: Climate warming over the past half century has led to thermal
degradation of permafrost on the Qinghai–Tibet Plateau. In: The Cryosphere, 12, 595–608

Thanks. Many recent articles have pointed out that the Qinghai-Tibet Plateau has warmed more than 0.25 degrees every ten years. We updated the sentence, and added a recent reference.

"The temperature in the QTP has increased by more than 0.25 °C per decade over the past 50 years (Li et al., 2010; Ran et al., 2018; Shen et al., 2015; Yao et al., 2007)."

Page 4, Line 5, I don't think this paragraph is necessary. This is a classical research article which structure

35    is well known and is already organised based on it. Also is indicated to finish the introduction part with the purpose, for a easier article reading.

We deleted this paragraph.

40    Page 5, Line 5, These indices are indeed very well explained.

We have added more descriptions to these permafrost indices in section "Permafrost modeling". We have updated the sentence as follows:

"DDT$_a$ and DDT$_s$ are the sums of mean daily air and ground surface above temperatures 0 °C (Celsius degree-days), respectively. DDF$_a$ and DDF$_s$ are the sums of mean daily air and ground surface temperatures below 0°C (Celsius degree-days), respectively."

"Local variations in vegetation, topography, and snow cover may result in several differences between MAGST and MAAT."

"MAGT is defined as the soil temperature at the depth of zero annual temperature change. T$_{z,t}$ is the ground temperature at any time t and depth z below a ground surface. MAGT is often found at depths from 10 m to 16 m over the QTP (Wu and Zhang, 2010), Here, we take the z value of 15 metres as default value, user can change the depth z."

"The seasonal thawing/freezing n factor (n$_t$/n$_f$) relates thawing and freezing degree-days (DDT$_a$/DDT$_s$/DDF$_a$/DDF$_s$) in seasonal air temperature to ground surface temperatures."

"TTOP indicates average temperatures at the top of the permafrost. The active layer is defined as the layer of ground subject to annual thawing and freezing underlain by permafrost."

Page 6, Line 16, Typing error. Subscript instead of normal letters.

This has been corrected, thank you.

Page 9, Line 3, It could be written also "Results" as a chapter name here. Or joined.

As a Development and technical manuscript of "Geoscientific Model Development", we reorganize the manuscript structure, and changed the title to "Implementation".

Page 10, Line 7-8, That's great, because is processing automatically the outliers.

Thanks.

Page 13, Line 2-3, Can this be explained?

The simulated TTOP and ALT that uses the Stefan and Smith functions have higher TTOP and ALT than the Kudryavtsev function. The difference between them were also shown in other areas (Uxa, 2017; Wilhelm et al., 2015).

Figure 1, Missing scale bar.
Maybe a bit more info on the map (r.g. main roads and rivers, key cities), and the dots for the weather stations can be smaller if it will be too crowded with the additional info.

Thanks for pointing this out. We added the scale, and also add lake, glacier (the Second Glacier Inventory Dataset of China, v1.0), the legend of elevation map, provincial border and three major cities in the Qinghai-Tibet Plateau.

[Figure]

Figure 1: Map of the data location over the QTP.

Table 2, Typing error: too much space.

This has been corrected, thank you.

Table 2, It could be a better matching of left columns with right columns in this part for a easier reading.

We adjusted the arrangement of the columns.

Table 2, column

This has been corrected, thank you.

Table 3, It could be a note under the table to mention the abbreviations or to indicate that the abbreviations are specified in text.

We have added the sentence in the table caption as follows:

"Intercept: y-intercept; Slope: slope of regression line; R: Pearson's correlation coefficient, R2: coefficient of determination; RMSE: root mean squared error; NRMSE: normalized RMSE; SD_S: the standard deviation of TTOP using the Stefan function; SD_K: the standard deviation of TTOP using the Kudryavtsev function; MEF: modelling efficiency; NAE: normalized average error; VR: variance ratio; PBIAS: percent bias; NSE: Nash-Sutchliffe efficiency; RSR: RMSE-observations standard deviation ratio; and D: index of agreement."

[revised manuscript text omitted]

---

## Author Response (AR2)

[revised manuscript text omitted]

